# Simple steps are all you need: Frank-Wolfe and generalized self-concordant functions

Alejandro Carderera [*]   Mathieu Besançon [†]   Sebastian Pokutta [‡]

## Abstract

Generalized self-concordance is a key property present in the objective function of many important learning problems. We establish the convergence rate of a simple Frank-Wolfe variant that uses the open-loop step size strategy $\gamma_t = 2/(t+2)$, obtaining a $O(1/t)$ convergence rate for this class of functions in terms of primal gap and Frank-Wolfe gap, where $t$ is the iteration count. This avoids the use of second-order information or the need to estimate local smoothness parameters of previous work. We also show improved convergence rates for various common cases, e.g., when the feasible region under consideration is uniformly convex or polyhedral.

## 1 Introduction

Constrained convex optimization is the cornerstone of many machine learning problems. We consider such problems, formulated as:

$$\min_{\mathbf{x} \in \mathcal{X}} f(\mathbf{x}), \tag{1.1}$$

where $f : \mathbb{R}^n \to \mathbb{R} \cup \{+\infty\}$ is a generalized self-concordant function and $\mathcal{X} \subseteq \mathbb{R}^n$ is a compact convex set. When computing projections onto the feasible regions as required in, e.g., projected gradient descent, is prohibitive, Frank-Wolfe (FW) (Frank & Wolfe, 1956) algorithms (a.k.a. Conditional Gradients (CG) (Levitin & Polyak, 1966)) are the algorithms of choice, relying on Linear Minimization Oracles (LMO) at each iteration to solve Problem (1.1). The analysis of their convergence often relies on the assumption that the gradient is Lipschitz-continuous. This assumption does not necessarily hold for generalized self-concordant functions, an important class of functions for which the growth can be unbounded.

### 1.1 Related work

In the classical analysis of Newton's method, when the Hessian of $f$ is assumed to be Lipschitz continuous and the function is strongly convex, one arrives at a convergence rate for the algorithm that depends on the Euclidean structure of $\mathbb{R}^n$, despite the fact that the algorithm is affine-invariant. This motivated the introduction of self-concordant functions in Nesterov & Nemirovskii (1994), functions for which the third derivative is bounded by the second-order derivative, with which one can obtain an affine-invariant convergence rate for the aforementioned algorithm. More importantly, many of the barrier functions used in interior-point methods are self-concordant, which extended the use of polynomial-time interior-point methods to many settings of interest.

[*]Georgia Institute of Technology. E-mail: alejandro.carderera@gatech.edu
[†]Zuse Institute Berlin. E-mail: besancon@zib.de
[‡]Zuse Institute Berlin and Technische Universität Berlin. E-mail: pokutta@math.tu-berlin.de

35th Conference on Neural Information Processing Systems (NeurIPS 2021).

Self-concordant functions have received strong interest in recent years due to the attractive properties that they allow to prove for many statistical estimation settings (Marteau-Ferey et al., 2019; Ostrovskii & Bach, 2021). The original definition of self-concordance has been expanded and generalized since its inception, as many objective functions of interest have self-concordant-like properties without satisfying the strict definition of self-concordance. For example, the logistic loss function used in logistic regression is not strictly self-concordant, but it fits into a class of pseudo-self-concordant functions, which allows one to obtain similar properties and bounds as those obtained for self-concordant functions (Bach, 2010). This was also the case in Ostrovskii & Bach (2021) and Tran-Dinh et al. (2015), in which more general properties of these pseudo-self-concordant functions were established. This was fully formalized in Sun & Tran-Dinh (2019), in which the concept of *generalized self-concordant* functions was introduced, along with key bounds, properties, and variants of Newton methods for the unconstrained setting which make use of this property.

Most algorithms that aim to solve Problem (1.1) assume access to second-order information, as this often allows the algorithms to make monotonous progress, remain inside the domain of $f$, and often, converge quadratically when close enough to the optimum. Recently, several lines of work have focused on using Frank-Wolfe algorithm variants to solve these types of problems in the projection-free setting, for example constructing second-order approximations to a self-concordant $f$ using first and second-order information, and minimizing these approximations over $\mathcal{X}$ using the Frank-Wolfe algorithm (Liu et al., 2020). Other approaches, such as the ones presented in Dvurechensky et al. (2020a) (later extended in Dvurechensky et al. (2020b)), apply the Frank-Wolfe algorithm to a generalized self-concordant $f$, using first and second-order information about the function to guarantee that the step sizes are so that the iterates do not leave the domain of $f$, and monotonous progress is made. An additional Frank-Wolfe variant in that work, in the spirit of Garber & Hazan (2016), utilizes first and second order information about $f$, along with a Local Linear Optimization Oracle for $\mathcal{X}$, to obtain a linear convergence rate in primal gap over polytopes given in inequality description. The authors in Dvurechensky et al. (2020b) also present an additional Frank-Wolfe variant which does not use second-order information, and uses the backtracking line search of Pedregosa et al. (2020) to estimate local smoothness parameters at a given iterate. Other specialized Frank-Wolfe algorithms have been developed for specific problems involving generalized self-concordant functions, such as the Frank-Wolfe variant developed for marginal inference with concave maximization (Krishnan et al., 2015), the variant developed in Zhao & Freund (2020) for $\theta$-homogeneous barrier functions, or the application for phase retrieval in Odor et al. (2016), where the Frank-Wolfe algorithm is shown to converge on a self-concordant non-Lipschitz smooth objective.

## 1.2 Contribution

The contributions of this paper are detailed below and summarized in Table 1.

**Simple FW for generalized self-concordant functions.** We show that a small variation of the original Frank-Wolfe algorithm (Frank & Wolfe, 1956) with an open-loop step size of the form $\gamma_t = 2/(t+2)$, where $t$ is the iteration count is all that is needed to achieve a convergence rate of $O(1/t)$ in primal gap; this also answers an open question posed in Dvurechensky et al. (2020b). Our variation ensures monotonous progress while employing an open-loop strategy which, together with the iterates being convex combinations, ensures that we do not leave the domain of $f$. In contrast to other methods that depend on either a line search or second-order information, our variant uses only a linear minimization oracle, zeroth-order and first-order information and a domain oracle for $f(\mathbf{x})$. The assumption of the latter oracle is very mild and was also implicitly assumed in several of the algorithms presented in Dvurechensky et al. (2020b). As such, our iterations are much cheaper than those in previous work, while essentially achieving the same convergence rates for Problem (1.1).

Moreover, our variant relying on the open-loop step size $\gamma_t = 2/(t+2)$ allows us to establish a $O(1/t)$ convergence rate for the Frank-Wolfe gap, is agnostic, i.e., does not need to estimate local smoothness parameters, and is parameter-free, leading to convergence rates and oracle complexities that are independent of any tuning parameters.

| Algorithm | Convergence | | Reference | $1^{\text{st}}$-order / LS free? | Requirements |
|---|---|---|---|---|---|
| | Primal gap | FW gap | | | |
| FW-GSC | $O(1/\varepsilon)$ | | [1, Alg.2] | ✗ / ✓ | SOO |
| LBTFW-GSC | $O(1/\varepsilon)$ | | [1, Alg.3] | ✓ / ✗ | ZOO, DO |
| MBTFW-GSC | $O(1/\varepsilon)$ | | [1, Alg.5] | ✗ / ✓ | ZOO, SOO, DO |
| FW-LLOO | $O(\log 1/\varepsilon)$ | | [1, Alg.7] | ✗ / ✓ | polyh. $\mathcal{X}$, LLOO, SOO |
| ASFW-GSC | $O(\log 1/\varepsilon)$ | | [1, Alg.8] | ✗ / ✓ | polyh. $\mathcal{X}$, SOO |
| M-FW | $O(1/\varepsilon)$ | $O(1/\varepsilon)$ | **This work** | ✓ / ✓ | ZOO,DO |
| B-AFW | $O(\log 1/\varepsilon)$ | $O(\log 1/\varepsilon)$ | **This work** | ✓ / ✗ | polyh. $\mathcal{X}$, ZOO,DO |

Table 1: Number of iterations needed to achieve an $\varepsilon$-optimal solution for Problem 1.1. We denote Dvurechensky et al. (2020b) by [1], line search by LS, local linear optimization oracle by LLOO, and the assumption that $\mathcal{X}$ is polyhedral by polyh. $\mathcal{X}$. The oracles listed under **Requirements** are the additional oracles needed, other than the FOO and the LMO.

**Faster rates in common special cases.** We also obtain improved convergence rates when the optimum is contained in the interior of $\mathcal{X} \cap \text{dom}(f)$, or when the set $\mathcal{X}$ is uniformly or strongly convex, using the backtracking line search of Pedregosa et al. (2020). We also show that the Away-Step Frank-Wolfe algorithm (Lacoste-Julien & Jaggi, 2015; Wolfe, 1970) can use the aforementioned line search to achieve linear rates over polytopes. For clarity we want to stress that any linear rate over polytopes has to depend also on the ambient dimension of the polytope; this applies to our linear rates and those in Table 1 established elsewhere (see Diakonikolas et al. (2020)). In contrast the $O(1/\varepsilon)$ rates are dimension-independent.

**Numerical experiments.** We provide numerical experiments that showcase the performance of the algorithms on generalized self-concordant objectives to complement the theoretical results. In particular, they highlight that the simple step size strategy we propose is competitive with and sometimes outperforms other variants on many instances.

After publication of our initial draft, in a revision of their original work, Dvurechensky et al. (2020b) added an analysis of the Away-Step Frank-Wolfe algorithm which is complementary to ours (considering a slightly different setup and regimes) and was conducted independently; we have updated the tables to include these additional results.

## 1.3 Preliminaries and Notation

We denote the *domain* of $f$ as $\text{dom}(f) \overset{\text{def}}{=} \{\mathbf{x} \in \mathbb{R}^n, f(\mathbf{x}) < +\infty\}$ and the (potentially non-unique) minimizer of Problem (1.1) by $\mathbf{x}^*$. Moreover, we denote the *primal gap* and the *Frank-Wolfe gap* at $\mathbf{x} \in \mathcal{X} \cap \text{dom}(f)$ as $h(\mathbf{x}) \overset{\text{def}}{=} f(\mathbf{x}) - f(\mathbf{x}^*)$ and $g(\mathbf{x}) \overset{\text{def}}{=} \max_{\mathbf{v} \in \mathcal{X}} \langle \nabla f(\mathbf{x}), \mathbf{x} - \mathbf{v} \rangle$, respectively. We use $\|\cdot\|$, $\|\cdot\|_H$, and $\langle \cdot, \cdot \rangle$ to denote the *Euclidean norm*, the *matrix norm* induced by a symmetric positive definite matrix $H \in \mathbb{R}^{n \times n}$, and the *Euclidean inner product*, respectively. We denote the *diameter* of $\mathcal{X}$ as $D \overset{\text{def}}{=} \max_{\mathbf{x},\mathbf{y} \in \mathcal{X}} \|\mathbf{x} - \mathbf{y}\|$. Given a non-empty set $\mathcal{X} \subset \mathbb{R}^n$ we refer to its *boundary* as $\text{Bd}(\mathcal{X})$ and to its *interior* as $\text{Int}(\mathcal{X})$. We use $\Delta_n$ to denote the *probability simplex* of dimension $n$. Given a compact convex set $C \subseteq \text{dom}(f)$ we denote $L_f^C = \max_{\mathbf{u} \in C, \mathbf{d} \in \mathbb{R}^n} \|\mathbf{d}\|_{\nabla^2 f(\mathbf{u})}^2 / \|\mathbf{d}\|_2^2$ and $\mu_f^C = \min_{\mathbf{u} \in C, \mathbf{d} \in \mathbb{R}^n} \|\mathbf{d}\|_{\nabla^2 f(\mathbf{u})}^2 / \|\mathbf{d}\|_2^2$. We assume access to:

1. **Domain Oracle (DO):** Given $\mathbf{x} \in \mathcal{X}$, return true if $\mathbf{x} \in \text{dom}(f)$, false otherwise.
2. **Zeroth-Order Oracle (ZOO):** Given $\mathbf{x} \in \text{dom}(f)$, return $f(\mathbf{x})$.
3. **First-Order Oracle (FOO):** Given $\mathbf{x} \in \text{dom}(f)$, return $\nabla f(\mathbf{x})$.
4. **Linear Minimization Oracle (LMO):** Given $\mathbf{d} \in \mathbb{R}^n$, return $\text{argmin}_{\mathbf{x} \in \mathcal{X}} \langle \mathbf{x}, \mathbf{d} \rangle$.

The FOO and LMO oracles are standard in the FW literature. The ZOO oracle is often implicitly assumed to be included with the FOO oracle; we make this explicit here for clarity. Finally, the DO oracle is motivated by the properties of generalized self-concordant functions. It is reasonable to assume the availability of the DO oracle: following the definition of the function codomain, one could simply evaluate $f$ at $\mathbf{x}$ and assert $f(\mathbf{x}) < +\infty$, thereby combining the DO and ZOO oracles into one oracle. However, in many cases testing the membership of $\mathbf{x} \in \text{dom}(f)$ is computationally less demanding than the function evaluation.

**Remark 1.1.** Requiring access to a zeroth-order and domain oracle are mild assumptions, that were also implicitly assumed in one of the three FW-variants presented in Dvurechensky et al. (2020b) when computing the step size according to the strategy from Pedregosa et al. (2020); see Line 3 in Algorithm 6 in the Appendix. The remaining two variants ensure that $\mathbf{x} \in \text{dom}(f)$ by using second-order information about $f$, which we explicitly do not rely on.

The following example motivates the use of Frank-Wolfe algorithms in the context of generalized self-concordant functions. We present more examples in the computational results.

**Example 1.2** (Intersection of a convex set with a polytope)**.** Consider Problem (1.1) where $\mathcal{X} = \mathcal{P} \cap C$, $\mathcal{P}$ is a polytope over which we can minimize a linear function efficiently, and $C$ is a convex compact set for which one can easily build a barrier function.

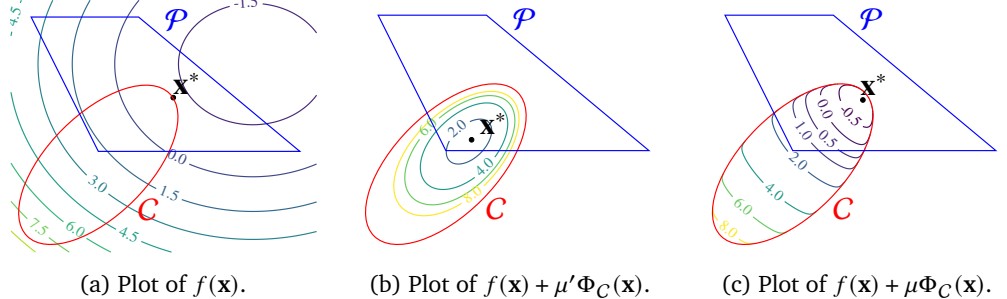

(a) Plot of $f(\mathbf{x})$.     (b) Plot of $f(\mathbf{x}) + \mu'\Phi_C(\mathbf{x})$.     (c) Plot of $f(\mathbf{x}) + \mu\Phi_C(\mathbf{x})$.

Figure 1: Minimizing $f(\mathbf{x})$ over $\mathcal{P} \cap C$, versus minimizing the sum of $f(\mathbf{x})$ and $\Phi_C(\mathbf{x})$ over $\mathcal{P}$ for two different penalty values $\mu'$ and $\mu$ such that $\mu' \gg \mu$.

Solving a linear optimization problem over $\mathcal{X}$ may be extremely expensive. In light of this, we can incorporate $C$ into the problem through the use of a barrier penalty in the objective function, minimizing instead $f(\mathbf{x}) + \mu\Phi_C(\mathbf{x})$ where $\Phi_C(\mathbf{x})$ is a log-barrier function for $C$ and $\mu$ is a parameter controlling the penalization. This reformulation is illustrated in Figure 1. Note that if the original objective function is generalized self-concordant, so is the new objective function (see Proposition 1 in Sun & Tran-Dinh (2019)). We assume that computing the gradient of $f(\mathbf{x}) + \mu\Phi_C(\mathbf{x})$ is roughly as expensive as computing the gradient for $f(\mathbf{x})$ and solving an LP over $\mathcal{P}$ is inexpensive relative to solving an LP over $\mathcal{P} \cap C$. The $\mu$ parameter can be driven down to 0 after a solution converges in a warm-starting procedure similar to interior-point methods, ensuring convergence to the true optimum.

An additional advantage of this transformation of the problem is the solution structure. Running Frank-Wolfe on the set $\mathcal{P} \cap C$ could potentially select a large number of extremal points from $\text{Bd}(C)$ if $C$ is non-polyhedral. In contrast, $\mathcal{P}$ has a finite number of vertices, a small subset of which will be selected throughout the optimization procedure. The same solution as that of the original problem can thus be constructed as a convex combination of a small number of vertices of $\mathcal{P}$, improving sparsity and interpretability in many applications.

The following definition formalizes the setting of Problem (1.1).

**Definition 1.3** (Generalized self-concordant function)**.** Let $f \in C^3(\text{dom}(f))$ be a closed convex function with $\text{dom}(f) \subseteq \mathbb{R}^n$ open. Then $f$ is $(M, \nu)$ *generalized self-concordant* if:

$$|\langle \mathrm{D}^3 f(\mathbf{x})[\mathbf{w}]\mathbf{u}, \mathbf{u}\rangle| \leq M \|\mathbf{u}\|_{\nabla^2 f(\mathbf{x})}^2 \|\mathbf{w}\|_{\nabla^2 f(\mathbf{x})}^{\nu-2} \|\mathbf{w}\|_2^{3-\nu},$$

for any $\mathbf{x} \in \text{dom}(f)$ and $\mathbf{u}, \mathbf{w} \in \mathbb{R}^n$, where $\mathrm{D}^3 f(\mathbf{x})[\mathbf{w}] = \lim_{\alpha \to 0} \alpha^{-1} \left(\nabla^2 f(\mathbf{x} + \alpha\mathbf{w}) - \nabla^2 f(\mathbf{x})\right)$.

## 2   Frank-Wolfe Convergence Guarantees

We establish convergence rates for a Frank-Wolfe variant with an open-loop step size strategy on generalized self-concordant functions. The *Monotonous Frank-Wolfe (M-FW)* algorithm presented in Algorithm 1 is a simple, but powerful modification of the standard Frank-Wolfe algorithm, with the only difference that before taking a step, we verify if $\mathbf{x}_t + \gamma_t(\mathbf{v}_t - \mathbf{x}_t) \in \text{dom}(f)$, and if so, we check whether moving to the next iterate provides primal progress. Note, that the open-loop step size rule $2/(t+2)$ does not guarantee monotonous primal

progress for the vanilla Frank-Wolfe algorithm in general. If either of these two checks fails, we do not move: the algorithm sets $\mathbf{x}_{t+1} = \mathbf{x}_t$ in Line 6 of Algorithm 1.

---

**Algorithm 1** Monotonous Frank-Wolfe (`M-FW`)

---

**Input:** Point $\mathbf{x}_0 \in \mathcal{X} \cap \mathrm{dom}(f)$, function $f$
**Output:** Iterates $\mathbf{x}_1, \ldots \in \mathcal{X}$

---

1: **for** $t = 0$ **to** $\ldots$ **do**
2: $\quad \mathbf{v}_t \leftarrow \mathrm{argmin}_{\mathbf{v} \in \mathcal{X}} \langle \nabla f(\mathbf{x}_t), \mathbf{v} \rangle$
3: $\quad \gamma_t \leftarrow 2/(t+2)$
4: $\quad \mathbf{x}_{t+1} \leftarrow \mathbf{x}_t + \gamma_t (\mathbf{v}_t - \mathbf{x}_t)$
5: $\quad$ **if** $\mathbf{x}_{t+1} \notin \mathrm{dom}(f)$ **or** $f(\mathbf{x}_{t+1}) > f(\mathbf{x}_t)$ **then**
6: $\quad\quad \mathbf{x}_{t+1} \leftarrow \mathbf{x}_t$

---

As customary, we assume short-circuit evaluation of the logical conditions in Algorithm 1, i.e., if the first condition in Line 5 is true, then the second condition is not even checked, and the algorithm directly goes to Line 6. This minor modification of the vanilla Frank-Wolfe algorithm enables us to use the monotonicity of the iterates in the proofs to come, at the expense of at most one extra function evaluation per iteration. Note that if we set $\mathbf{x}_{t+1} = \mathbf{x}_t$, we do not need to call the FOO or LMO oracle at iteration $t + 1$, as we can simply reuse $\nabla f(\mathbf{x}_t)$ and $\mathbf{v}_t$. This effectively means that between successive iterations in which we search for an acceptable value of $\gamma_t$, we only need to call the zeroth-order and domain oracle.

**Remark 2.1.** In practice, one can search for a suitable step size logarithmically, that is, halving the value of $\gamma_t$ if either $\mathbf{x}_{t+1} \notin \mathrm{dom}(f)$ or $f(\mathbf{x}_{t+1}) > f(\mathbf{x}_t)$. This would lead to a step size that is at most a factor of 2 smaller than the non-zero step size that would have been eventually accepted by the Monotonous Frank-Wolfe (`M-FW`) algorithm in Algorithm 1. Moreover, this strategy would allow us to obtain convergence rates that are very similar to those of the Monotonous Frank-Wolfe (`M-FW`) algorithm, with the exception of a factor of 2. The motivation and implications of this variant are detailed in Appendix A. For simplicity, we focus on the step size presented in Algorithm 1 in the main body of the paper.

In order to establish the main convergence results for the algorithm, we lower bound the progress per iteration with the help of Proposition 2.2.

**Proposition 2.2.** *(C.f., (Sun & Tran-Dinh, 2019, Proposition 10)) Given a $(M, \nu)$ generalized self-concordant function, then for $\nu \geq 2$, we have that:*

$$f(\mathbf{y}) - f(\mathbf{x}) - \langle \nabla f(\mathbf{x}), \mathbf{y} - \mathbf{x} \rangle \leq \omega_\nu(d_\nu(\mathbf{x} - \mathbf{y})) \|\mathbf{y} - \mathbf{x}\|^2_{\nabla^2 f(\mathbf{x})}, \tag{2.1}$$

*where the inequality holds if and only if $d_\nu(\mathbf{x}, \mathbf{y}) < 1$ for $\nu > 2$, and we have that,*

$$d_\nu(\mathbf{x}, \mathbf{y}) \overset{\text{def}}{=} \begin{cases} M \|\mathbf{y} - \mathbf{x}\| & \text{if } \nu = 2 \\ (\frac{\nu}{2} - 1) M \|\mathbf{y} - \mathbf{x}\|^{3-\nu} \|\mathbf{y} - \mathbf{x}\|^{\nu-2}_{\nabla^2 f(\mathbf{x})} & \text{if } \nu > 2, \end{cases}$$

*where:*

$$\omega_\nu(\tau) \overset{\text{def}}{=} \begin{cases} \frac{e^\tau - \tau - 1}{\tau^2} & \text{if } \nu = 2 \\ \frac{-\tau - \ln(1-\tau)}{\tau^2} & \text{if } \nu = 3 \\ \frac{(1-\tau)\ln(1-\tau)+\tau}{\tau^2} & \text{if } \nu = 4 \\ \left(\frac{\nu-2}{4-\nu}\right)\frac{1}{\tau}\left[\frac{\nu-2}{2(3-\nu)\tau}\left((1-\tau)^{\frac{2(3-\nu)}{2-\nu}} - 1\right) - 1\right] & \text{otherwise.} \end{cases}$$

The inequality shown in Equation (2.1) is very similar to the one that we would obtain if the gradient of $f$ were Lipschitz continuous, however, while the Lipschitz continuity of the gradient leads to an inequality that holds globally for all $\mathbf{x}, \mathbf{y} \in \mathrm{dom}(f)$, the inequality in Equation (2.1) only holds for $d_\nu(\mathbf{x}, \mathbf{y}) < 1$. Moreover, there are two other important differences, the norm used in Equation (2.1) is now the norm defined by the Hessian at $\mathbf{x}_t$ instead of the $\ell_2$ norm, and the term multiplying the norm is $\omega_\nu(d_\nu(\mathbf{x}, \mathbf{y}))$ instead of $1/2$. We deal with the latter issue by bounding $\omega_\nu(d_\nu(\mathbf{x}, \mathbf{y}))$ with a constant that depends on $\nu$ for any $\mathbf{x}, \mathbf{y} \in \mathrm{dom}(f)$ such that $d_\nu(\mathbf{x}, \mathbf{y}) \leq 1/2$, as shown in Remark 2.3.

**Remark 2.3.** As $\mathrm{d}\,\omega_\nu(\tau)/\mathrm{d}\,\tau > 0$ for $\tau < 1$ and $\nu \geq 2$, then $\omega_\nu(\tau) \leq \omega_\nu(1/2)$ for $\tau \leq 1/2$.

Due to the fact that we use a simple step size $\gamma_t = 2/(t+2)$, that we make monotonous progress, and we ensure that the iterates are inside $\mathrm{dom}(f)$, careful accounting allows us to bound the number of iterations until $d_\nu(\mathbf{x}_t, \mathbf{x}_t + \gamma_t(\mathbf{v}_t - \mathbf{x}_t)) \leq 1/2$. Before formalizing the convergence rate we first review a lemma that we will need in the proof.

**Lemma 2.4.** *(C.f., (Sun & Tran-Dinh, 2019, Proposition 7)) Let $f$ be a generalized self-concordant function with $\nu > 2$. If $d_\nu(\mathbf{x}, \mathbf{y}) < 1$ and $\mathbf{x} \in \mathrm{dom}(f)$ then $\mathbf{y} \in \mathrm{dom}(f)$. For the case $\nu = 2$ we have that $\mathrm{dom}(f) = \mathbb{R}^n$.*

Putting all these things together allows us to obtain a convergence rate for Algorithm 1.

**Theorem 2.5.** *Suppose $X$ is a compact convex set and $f$ is a $(M, \nu)$ generalized self-concordant function with $\nu \geq 2$. Then the Monotonous Frank-Wolfe algorithm (Algorithm 1) satisfies:*

$$h(\mathbf{x}_t) \leq \frac{4(T_\nu + 1)}{t + 1} \max\left\{h(\mathbf{x}_0), L_f^{\mathcal{L}_0} D^2 \omega_\nu(1/2)\right\}. \tag{2.2}$$

*for $t \geq T_\nu$, where $L_f^{\mathcal{L}_0} = \max\limits_{\mathbf{u} \in \mathcal{L}_0, \mathbf{d} \in \mathbb{R}^n} \|\mathbf{d}\|^2_{\nabla^2 f(\mathbf{u})} / \|\mathbf{d}\|^2_2$ and $T_\nu$ is defined as:*

$$T_\nu \stackrel{\mathrm{def}}{=} \begin{cases} \lceil 4MD \rceil - 2 & \text{if } \nu = 2 \\ \left\lceil 2MD(L_f^{\mathcal{L}_0})^{\nu/2-1}(\nu - 2) \right\rceil - 2 & \text{otherwise.} \end{cases} \tag{2.3}$$

*Otherwise it holds that $h(\mathbf{x}_t) \leq h(\mathbf{x}_0)$ for $t < T_\nu$.*

*Proof.* Consider the compact set $\mathcal{L}_0 \stackrel{\mathrm{def}}{=} \{\mathbf{x} \in \mathrm{dom}(f) \cap X \mid f(\mathbf{x}) \leq f(\mathbf{x}_0)\}$. As the algorithm makes monotonous progress and moves towards points such that $\mathbf{x}_t \in X$, then $\mathbf{x}_t \in \mathcal{L}_0$ for $t \geq 0$. As the smoothness parameter of $f$ is bounded over $\mathcal{L}_0$, we have from the properties of smooth functions that the bound $\|\mathbf{d}\|^2_{\nabla^2 f(\mathbf{x}_t)} / \|\mathbf{d}\|^2 \leq L_f^{\mathcal{L}_0}$ holds for any $\mathbf{d} \in \mathbb{R}^n$. Particularizing for $\mathbf{d} = \mathbf{x}_t - \mathbf{v}_t$ and noting that $\|\mathbf{x}_t - \mathbf{v}_t\| \leq D$ leads to $\|\mathbf{x}_t - \mathbf{v}_t\|^2_{\nabla^2 f(\mathbf{x}_t)} \leq L_f^{\mathcal{L}_0} D^2$. We then define $T_\nu$ as in Equation (2.3). Note that for $t \geq T_\nu$ we have that $d(\mathbf{x}_t, \mathbf{x}_t + \gamma_t(\mathbf{v}_t - \mathbf{x}_t)) \leq 1/2$, and so as $\mathbf{x}_t \in \mathrm{dom}(f)$ we will have $\mathbf{x}_t + \gamma_t(\mathbf{v}_t - \mathbf{x}_t) \in \mathrm{dom}(f)$, by application of Lemma 2.4. This means that the non-zero step size $\gamma_t$ will ensure that $\mathbf{x}_t + \gamma_t(\mathbf{v}_t - \mathbf{x}_t) \in \mathrm{dom}(f)$ in Line 5 of Algorithm 1. Moreover, it allows us to use the bound between points $\mathbf{x}_t$ and $\mathbf{x}_t + \gamma_t(\mathbf{v}_t - \mathbf{x}_t)$ in Proposition 2.2, which holds for $d(\mathbf{x}_t, \mathbf{x}_t + \gamma_t(\mathbf{v}_t - \mathbf{x}_t)) < 1$. With this we can estimate the primal progress we can guarantee for $t \geq T_\nu$ if we move from $\mathbf{x}_t$ to $\mathbf{x}_t + \gamma_t(\mathbf{v}_t - \mathbf{x}_t)$:

$$h(\mathbf{x}_t + \gamma_t(\mathbf{v}_t - \mathbf{x}_t)) \leq h(\mathbf{x}_t) - \gamma_t g(\mathbf{x}_t) + \gamma_t^2 \omega_\nu(d_\nu(\mathbf{x}_t, \mathbf{x}_t + \gamma_t(\mathbf{v}_t - \mathbf{x}_t))) \|\mathbf{v}_t - \mathbf{x}_t\|^2_{\nabla^2 f(\mathbf{x}_t)}$$

$$\leq h(\mathbf{x}_t)(1 - \gamma_t) + \gamma_t^2 L_f^{\mathcal{L}_0} D^2 \omega_\nu(1/2),$$

where the second inequality follows from the upper bound on the primal gap via the Frank-Wolfe gap $g(\mathbf{x}_t)$, the application of Remark 2.3 as for $t \geq T_\nu$ we have that $d_\nu(\mathbf{x}_t, \mathbf{x}_t + \gamma_t(\mathbf{v}_t - \mathbf{x}_t)) \leq 1/2$, and from the fact that $\mathbf{x}_t \in \mathcal{L}_0$ for all $t \geq 0$. With the previous chain of inequalities we can bound the primal progress for $t \geq T_\nu$ as

$$h(\mathbf{x}_t) - h(\mathbf{x}_t + \gamma_t(\mathbf{v}_t - \mathbf{x}_t)) \geq \gamma_t h(\mathbf{x}_t) - \gamma_t^2 L_f^{\mathcal{L}_0} D^2 \omega_\nu(1/2). \tag{2.4}$$

From these facts we can prove the convergence rate shown in Equation (2.2) by induction. The base case $t = T_\nu$ holds trivially by the fact that using monotonicity we have that $h(\mathbf{x}_{T_\nu}) \leq h(\mathbf{x}_0)$. Assuming the claim is true for some $t \geq T_\nu$ we distinguish two cases.

**Case $\gamma_t h(\mathbf{x}_t) - \gamma_t^2 L_f^{\mathcal{L}_0} D^2 \omega_\nu(1/2) > 0$:** Focusing on the first case, we can plug the previous inequality into Equation (2.4) to find that $\gamma_t$ guarantees primal progress, that is, $h(\mathbf{x}_t) > h(\mathbf{x}_t + \gamma_t(\mathbf{v}_t - \mathbf{x}_t))$ with the step size $\gamma_t$, and so we know that we will not go into Line 6 of Algorithm 1, and we have that $h(\mathbf{x}_{t+1}) = h(\mathbf{x}_t + \gamma_t(\mathbf{v}_t - \mathbf{x}_t))$. Thus using the induction hypothesis and plugging in the expression for $\gamma_t = 2/(t + 2)$ into Equation (2.4) we have:

$$h(\mathbf{x}_{t+1}) \leq 4 \max\left\{h(\mathbf{x}_0), L_f^{\mathcal{L}_0} D^2 \omega_\nu(1/2)\right\} \left(\frac{(T_\nu + 1)t}{(t + 1)(t + 2)} + \frac{1}{(t + 2)^2}\right)$$

$$\leq \frac{4(T_\nu + 1)}{t + 2} \max\left\{h(\mathbf{x}_0), L_f^{\mathcal{L}_0} D^2 \omega_\nu(1/2)\right\},$$

where we use that $(T_\nu + 1)t/(t + 1) + 1/(t + 2) \leq T_\nu + 1$ for all $t \geq 0$ and any $t \geq T_\nu$.

**Case $\gamma_t h(\mathbf{x}_t) - \gamma_t^2 L_f^{\mathcal{L}_0} D^2 \omega_\nu(1/2) \leq 0$:** In this case, we cannot guarantee that the step size $\gamma_t$ provides primal progress by plugging into Equation (2.4), and so we cannot guarantee if a step size of $\gamma_t$ will be accepted and we will have $\mathbf{x}_{t+1} = \mathbf{x}_t + \gamma_t(\mathbf{v}_t - \mathbf{x}_t)$, or we will simply

have $\mathbf{x}_{t+1} = \mathbf{x}_t$, that is, we may go into Line 6 of Algorithm 1. Nevertheless, if we reorganize the expression $\gamma_t h(\mathbf{x}_t) - \gamma_t^2 L_f^{\mathcal{L}_0} D^2 \omega_\nu(1/2) \le 0$, by monotonicity we will have that:

$$h(\mathbf{x}_{t+1}) \le h(\mathbf{x}_t) \le \frac{2}{t+2} L_f^{\mathcal{L}_0} D^2 \omega_\nu(1/2) \le \frac{4(T_\nu + 1)}{t+2} \max\left\{ h(\mathbf{x}_0), L_f^{\mathcal{L}_0} D^2 \omega_\nu(1/2) \right\}.$$

Where the last inequality holds as $2 \le 4(T_\nu + 1)$ for any $T_\nu \ge 0$. □

**Remark 2.6.** In the case where $\nu = 2$ we can easily bound the primal gap $h(\mathbf{x}_1)$, as in this setting $\text{dom}(f) = \mathbb{R}^n$, which leads to $h(\mathbf{x}_1) \le L_f^{\mathcal{X}} D^2$ from Equation (2.4), regardless of if we set $\mathbf{x}_1 = \mathbf{x}_0$ or $\mathbf{x}_1 = \mathbf{v}_0$. Moreover, as the upper bound on the Bregman divergence holds for $\nu = 2$ regardless of the value of $d_2(\mathbf{x}, \mathbf{y})$, we can modify the proof of Theorem 2.5 to obtain a convergence rate of the form $h(\mathbf{x}_t) \le 2/(t+1) L_f^{\mathcal{X}} D^2 w_2(MD)$ for $t \ge 1$, which is reminiscent of the $O(L_f^{\mathcal{X}} D^2/t)$ rate of the original Frank-Wolfe algorithm for the smooth and convex case.

Note that in the proof of Theorem 2.5 we explicitly use the progress bound from generalized self-concordance as opposed to the progress bound that arises from $L_f^{\mathcal{L}_0}$-smoothness, as there is no straightforward way to bound the number of iterations until the latter progress bound holds indefinitely for all $\mathbf{x}_t + \gamma_t(\mathbf{v}_t - \mathbf{x}_t)$, while there is a straightforward criterion on $\gamma_t$ that allows us to ensure that the former holds from some point onward (see Remark A.1 in the Appendix for more details). Furthermore, with this simple step size we can also prove a convergence rate for the Frank-Wolfe gap, as shown in Theorem 2.7 (see Theorem A.2 in the Appendix for the proof).

**Theorem 2.7.** *Suppose $\mathcal{X}$ is a compact convex set and $f$ is a $(M, \nu)$ generalized self-concordant function with $\nu \ge 2$. Then if the Monotonous Frank-Wolfe algorithm (Algorithm 1) is run for $T \ge T_\nu + 6$ iterations, we will have that $\min_{1 \le t \le T} g(\mathbf{x}_t) \le O(1/T)$.*

**Remark 2.8.** Note that the Monotonous Frank-Wolfe algorithm (Algorithm 1) performs at most one ZOO, FOO, DO, and LMO oracle call per iteration. This means that Theorems 2.5 and 2.7 effectively bound the number of ZOO, FOO, DO, and LMO oracle calls needed to achieve a target primal gap or Frank-Wolfe gap accuracy $\varepsilon$ as a function of $T_\nu$ and $\varepsilon$; note that $T_\nu$ is independent of $\varepsilon$. This is an important difference with respect to existing bounds, as the existing Frank-Wolfe-style first-order algorithms for generalized self-concordant functions in the literature that utilize various types of line searches may perform more than one ZOO or DO call per iteration in the line search. This means that the convergence bounds in terms of iteration count of these algorithms are only informative when considering the number of FOO and LMO calls that are needed to reach a target accuracy in primal gap, and do not directly provide any information regarding the number of ZOO or DO calls that are needed. In order to bound the latter two quantities one typically needs additional technical tools. For example, for the backtracking line search of Pedregosa et al. (2020), one can use Theorem 1 in Appendix C of Pedregosa et al. (2020), or a slightly modified version of Lemma 4 in Nesterov (2013), to find a bound for the number of ZOO or DO calls that are needed to find an $\varepsilon$-optimal solution. Note that these bounds depend on user-defined initialization or tuning parameters provided by the user.

In Table 2 we provide a detailed complexity comparison between the Monotonous Frank-Wolfe (M-FW) algorithm (Algorithm 1), and other comparable algorithms in the literature.

| Algorithm | SOO calls | FOO calls | ZOO calls | LMO calls | DO calls |
|---|---|---|---|---|---|
| FW-GSC [1, Alg.2] | $O(1/\varepsilon)$ | $O(1/\varepsilon)$ | | $O(1/\varepsilon)$ | |
| LBTFW-GSC$^\ddagger$ [1, Alg.3] | | $O(1/\varepsilon)$ | $O(1/\varepsilon)$ | $O(1/\varepsilon)$ | $O(1/\varepsilon)$ |
| MBTFW-GSC$^\ddagger$ [1, Alg.5] | $O(1/\varepsilon)$ | $O(1/\varepsilon)$ | $O(1/\varepsilon)$ | $O(1/\varepsilon)$ | $O(1/\varepsilon)$ |
| M-FW$^\dagger$ [**This work**] | | $O(1/\varepsilon)$ | $O(1/\varepsilon)$ | $O(1/\varepsilon)$ | $O(1/\varepsilon)$ |

Table 2: **Complexity comparison**: Number of iterations needed to reach a solution with $h(\mathbf{x})$ below $\varepsilon$ for Problem 1.1. We denote Dvurechensky et al. (2020b) using [1]. We use the superscript † to indicate that the same complexities hold for reaching an $\varepsilon$-optimal solution in $g(\mathbf{x})$. The superscript ‡ is used to indicate that constants in the convergence bounds depend on user-defined inputs; the other algorithms are parameter-free.

We note that the `LBTFW-GSC` algorithm from Dvurechensky et al. (2020b) is in essence the Frank-Wolfe algorithm with a modified version of the backtracking line search of Pedregosa et al. (2020). In the next section, we provide improved convergence guarantees for various cases of interest for this algorithm, which we refer to as the Frank-Wolfe algorithm with Backtrack (`B-FW`) for simplicity.

## 2.1 Improved convergence guarantees

**Algorithm 2** Frank-Wolfe with `Backtrack` of Pedregosa et al. (2020) (`B-FW`)

**Input:** $\mathbf{x}_0 \in \mathcal{X} \cap \text{dom}(f)$, function $f$, estimate $L_{-1}$
**Output:** Iterates $\mathbf{x}_1, \ldots \in \mathcal{X}$

1: **for** $t = 0$ **to** $\ldots$ **do**
2: $\quad \mathbf{v}_t \leftarrow \text{argmin}_{\mathbf{v} \in \mathcal{X}} \langle \nabla f(\mathbf{x}_t), \mathbf{v} \rangle$
3: $\quad \gamma_t, L_t \leftarrow \text{Backtrack}(f, \mathbf{x}_t, \mathbf{v}_t - \mathbf{x}_t, L_{t-1}, 1)$
4: $\quad \mathbf{x}_{t+1} \leftarrow \mathbf{x}_t + \gamma_t (\mathbf{v}_t - \mathbf{x}_t)$

We will now establish improved convergence rates for various special cases. We first focus on the assumption that $\mathbf{x}^* \in \text{Int}(\mathcal{X} \cap \text{dom}(f))$, obtaining improved rates when we use the FW algorithm coupled with the adaptive step size strategy from Pedregosa et al. (2020) (see Algorithm 6 in Appendix). This assumption is reasonable if for example $\text{Bd}(\mathcal{X}) \not\subseteq \text{dom}(f)$, and $\text{Int}(\mathcal{X}) \subseteq \text{dom}(f)$. That is to say, we will have that $\mathbf{x}^* \in \text{Int}(\mathcal{X} \cap \text{dom}(f))$ if for example we use logarithmic barrier functions to encode a set of constraint, and we have that $\text{dom}(f)$ is a proper subset of $\mathcal{X}$. In this case the optimum is guaranteed to be in $\text{Int}(\mathcal{X} \cap \text{dom}(f))$.

The analysis in this case is reminiscent of the one in the seminal work of Guélat & Marcotte (1986), and is presented in Appendix A.2. Note that we can upper bound the value of $L_t$ for $t \geq 0$ by $\tilde{L} \overset{\text{def}}{=} \max\{\tau L_f^{\mathcal{L}_0}, L_{-1}\}$, where $\tau > 1$ is the backtracking parameter and $L_{-1}$ is the initial smoothness estimate in Algorithm 6 .

**Theorem 2.9.** *Let $f$ be a $(M, \nu)$ generalized self-concordant function with $\nu \geq 2$ and let $\text{dom}(f)$ not contain straight lines. Furthermore, we denote by $r > 0$ the largest value such that $\mathcal{B}(\mathbf{x}^*, r) \subseteq \mathcal{X} \cap \text{dom}(f)$. Then the Frank-Wolfe algorithm with Backtrack (Algorithm 2) achieves a convergence rate for $t \geq 1$ of $h(\mathbf{x}_t) \leq h(\mathbf{x}_0) \left(1 - \mu_f^{\mathcal{L}_0}/(2\tilde{L})(r/D)^2\right)^t$.*

The assumption that $\text{dom}(f)$ does not contain straight lines in Theorem 2.9 is related to the Hessian being positive definite over $\text{dom}(f)$ (see the proof in the Appendix in Theorem A.5). Note that this is a very mild assumption as we can simply modify the function with a very small $\ell_2$ regularizer, as e.g., in Nesterov (2012). Next, we recall the definition of uniformly convex sets, used in Kerdreux et al. (2021), which will allow us to obtain improved convergence rates for the FW algorithm over uniformly convex feasible regions.

**Definition 2.10** $((\kappa, q)$-uniformly convex set$)$. *Given two positive numbers $\kappa$ and $q$, we say the set $\mathcal{X} \subseteq \mathbb{R}^n$ is $(\kappa, q)$-uniformly convex with respect to a norm $\|\cdot\|$ if for any $\mathbf{x}, \mathbf{y} \in \mathcal{X}$, $0 \leq \gamma \leq 1$, and $\mathbf{z} \in \mathbb{R}^n$ with $\|\mathbf{z}\| = 1$ we have that $\mathbf{y} + \gamma(\mathbf{x} - \mathbf{y}) + \gamma(1 - \gamma) \cdot \kappa \|\mathbf{x} - \mathbf{y}\|^q \mathbf{z} \in \mathcal{X}$.*

**Theorem 2.11.** *Suppose $\mathcal{X}$ is a compact $(\kappa, q)$-uniformly convex set and $f$ is a $(M, \nu)$ generalized self-concordant function with $\nu \geq 2$. Furthermore, assume that $\min_{\mathbf{x} \in \mathcal{X}} \|\nabla f(\mathbf{x})\| \geq C > 0$. Then the Frank-Wolfe algorithm with Backtrack (Algorithm 2) achieves a convergence rate of:*

$$
h_t \leq \begin{cases} h(\mathbf{x}_0) \left(1 - \frac{1}{2} \min\left\{1, \frac{\kappa C}{\tilde{L}}\right\}\right)^t & \text{if } q = 2 \\ \frac{h(\mathbf{x}_0)}{2^t} & \text{if } q > 2, 1 \leq t \leq t_0 \\ \frac{\tilde{L}^{q/(q-2)}/(\kappa C)^{2/(q-2)}}{(1+(q-2)(t-t_0)/(2q))^{q/(q-2)}} = O\left(t^{-q/(q-2)}\right) & \text{if } q > 2, t > t_0, \end{cases}
$$

*for $t \geq 1$, where $t_0 = \max\{1, \lfloor \log_{1/2}((\tilde{L}^q/(\kappa C)^2)^{1/(q-2)} h(\mathbf{x}_0)) \rfloor\}$.*

However, in the general case, we cannot assume that the norm of the gradient is bounded away from zero over $\mathcal{X}$. We deal with the general case in Theorem 2.12

**Theorem 2.12.** *Suppose $\mathcal{X}$ is a compact $(\kappa, q)$-uniformly convex set and $f$ is a $(M, \nu)$ generalized self-concordant function with $\nu \geq 2$ for which the domain does not contain straight lines. Then*

*the Frank-Wolfe algorithm with Backtrack (Algorithm 2) results in a convergence rate:*

$$h_t \leq \begin{cases} \frac{h(\mathbf{x}_0)}{2^t} & \text{if } 1 \leq t \leq t_0 \\ \frac{(\tilde{L}^q/(\kappa^2 \mu_f^{\mathcal{L}_0}))^{1/(q-1)}}{(1+(q-1)(t-t_0)/(2q))^{q/(q-1)}} = O\left(t^{-q/(q-1)}\right) & \text{if } t > t_0, \end{cases}$$

*for $t \geq 1$, where $t_0 = \max\{1, \lfloor \log_{1/2}((\tilde{L}^q/(\kappa^2 \mu_f^{\mathcal{L}_0}))^{1/(q-1)}/h(\mathbf{x}_0))\rfloor\}$.*

In Table 3 in Appendix A.2 we summarize the oracle complexity results shown in this paper for the `B-FW` algorithm when minimizing over a $(\kappa, q)$-uniformly convex set. Note that this algorithm is referred to as `LBTFW-GSC` in Dvurechensky et al. (2020b).

**Remark 2.13.** Contrary to previous claims, there is no obstacle for the *Away-step Frank-Wolfe* (AFW) algorithm (Guélat & Marcotte, 1986; Lacoste-Julien & Jaggi, 2015) together with the step size strategy in Algorithm 6 to obtain a linear convergence rate in primal and Frank-Wolfe gap when $\mathcal{X}$ is a polytope and $f$ is generalized self-concordant. This is not surprising, as $f$ is strongly convex and smooth over $\mathcal{L}_0$ if dom$(f)$ does not contain straight lines, and monotonicity ensures the feasibility of the iterates. We leave the analysis for this case to Appendix B, and the formal convergence statement to Theorem B.2 and B.3.

In Table 4 in Appendix B we provide a complexity comparison between the `B-AFW` algorithm, which can be found in Algorithm 7 in the appendix, and other comparable algorithms in the literature. Note that these complexities assume that $\mathcal{X}$ is polyhedral.

## 3 Computational experiments

We showcase the performance of the `M-FW` algorithm, the second-order step size and the LLOO algorithm from Dvurechensky et al. (2020b) (denoted by `GSC-FW` and `LLOO` in the figures) and the Frank-Wolfe and the Away-Step Frank-Wolfe algorithm with the backtracking stepsize of Pedregosa et al. (2020), denoted by `B-FW` and `B-AFW` respectively. All experiments are carried out in `Julia` using the `FrankWolfe.jl` package (Besançon et al., 2021), and the examples considered extend the ones presented in Dvurechensky et al. (2020b) and Liu et al. (2020). The code can be found in the `fw-generalized-selfconcordant` repository. We also use the vanilla FW algorithm denoted by `FW`, which is simply Algorithm 1 without Lines 5 and 6 using the traditional $\gamma_t = 2/(t+2)$ open-loop step size rule. Note that there are no formal convergence guarantees for this algorithm when applied to Problem (1.1). Details and remarks on the data and the experimental setup are provided in Appendix C. All figures show the evolution of the $h(\mathbf{x}_t)$ and $g(\mathbf{x}_t)$ against $t$ and time with a log-log scale. As in Dvurechensky et al. (2020b) we implemented the LLOO based variant only for the portfolio optimization instance $\Delta_n$; for the other examples, the oracle implementation was not implemented due to the need to estimate non-trivial parameters.

As can be seen in all experiments, the Monotonous Frank-Wolfe algorithm is very competitive, outperforming previously proposed variants in both in progress per iteration and time. The only other algorithm that is sometimes faster is the Away-Step Frank-Wolfe variant as detailed in Remark 2.13, which however depends on an active set, and can induce up to a quadratic overhead, making iterations progressively more expensive; this can also be observed in our experiments as the advantage in time is much less pronounced than in iterations.

**Portfolio optimization.** We consider $f(\mathbf{x}) = -\sum_{t=1}^{p} \log(\langle \mathbf{r}_t, \mathbf{x} \rangle)$, where $p$ denotes the number of periods and $\mathcal{X} = \Delta_n$. The results are shown in Figure 2.

**Signal recovery with KL divergence.** We apply the aforementioned algorithms to the recovery of a sparse signal from a noisy linear image using the Kullback-Leibler divergence, expressed as $f(\mathbf{x}) = D(W\mathbf{x}, \mathbf{y}) = \sum_{i=1}^{N} \left\{ \langle \mathbf{w}_i, \mathbf{x} \rangle \log \left( \frac{\langle \mathbf{w}_i, \mathbf{x} \rangle}{y_i} \right) - \langle \mathbf{w}_i, \mathbf{x} \rangle + y_i \right\}$, where $\mathbf{w}_i$ is the $i^{\text{th}}$ row of a matrix $W$. In order to promote sparsity and enforce nonnegativity of the solution, we use the unit simplex of radius $R$ as the feasible set $\mathcal{X} = \{\mathbf{x} \in \mathbb{R}_+^d, \|\mathbf{x}\|_1 \leq R\}$. The results are shown in Figure 3. We used the same $M = 1$ choice for the second-order method as in Dvurechensky et al. (2020b) for comparison; its admissibility is unknown (see Remark C.1).

**Logistic regression.** We consider a logistic regression task with a design matrix with rows $\mathbf{a}_i \in \mathbb{R}^n$ with $1 \le i \le N$ and a vector $\mathbf{y} \in \{-1, 1\}^N$ and formulate the problem with elastic net regularization, in a similar fashion as is done in Liu et al. (2020), with $f(\mathbf{x}) = 1/N \sum_{i=1}^{N} \log(1 + \exp(-y_i \langle \mathbf{x}, \mathbf{a}_i \rangle)) + \mu/2 \|\mathbf{x}\|^2$, where $\mu$ is a regularization parameter and $\mathcal{X}$ is the $\ell_1$ ball of radius $\rho$. The results can be seen in Figure 4 and Appendix C.

**Birkhoff polytope.** All previously considered applications have in common a feasible region possessing computationally inexpensive LMOs (probability/unit simplex and $\ell_1$ norm ball). Additionally, each vertex returned from the LMO is highly sparse with at most one non-zero element. To complement the results, we consider a convex quadratic optimization problem over the Birkhoff polytope, where the LMO call uses the Hungarian method and is not as inexpensive as in the other examples. The results are shown in Figure 5.

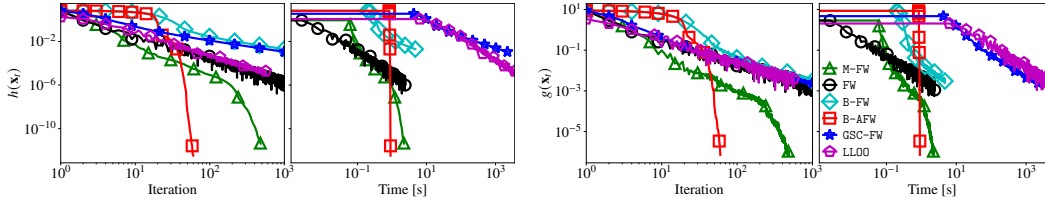

Figure 2: **Portfolio Optimization**: `LLOO` and `GSC-FW` perform similarly to `FW` on a per-iteration basis but the iterations are computationally more expensive. `B-AFW` is the fastest method both in terms iteration and runtime, followed by `M-FW` which is the only other method to terminate with the specified dual gap tolerance.

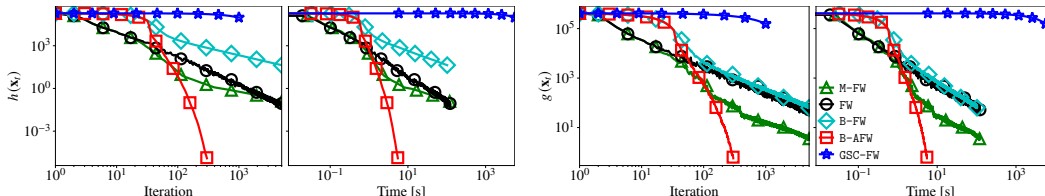

Figure 3: **Signal Recovery**: `B-AFW` significantly outperforms all other methods. `FW` and `B-FW` perform similarly in dual gap progress and converge slower than `M-FW`. In terms of primal gap progress, `M-FW` and `FW` perform similarly and outperform `B-FW`.

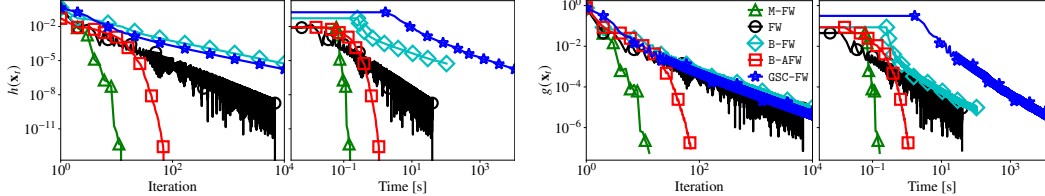

Figure 4: **Logistic Regression**: This instance shows that although simple in essence, `M-FW` can outperform other methods including `B-AFW` in terms of convergence. The primal and dual gaps for `B-FW` and `GSC-FW` converge at similar rates against iteration count.

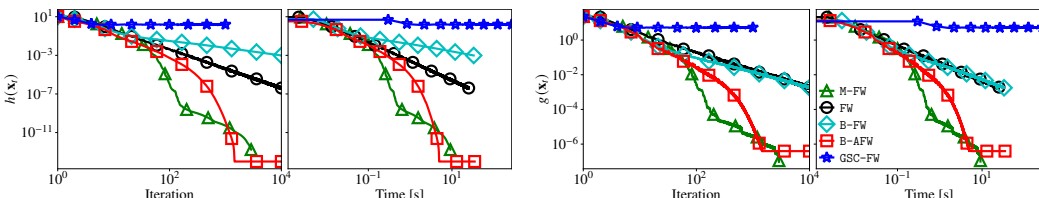

Figure 5: **Birkhoff Polytope**: `B-AFW` is the fastest-converging method for all measures. However, the dual gap reaches a plateau due to numerical issues above the termination threshold, unlike `M-FW` which reaches the dual gap tolerance. `GSC-FW` is run for 1000 iterations only given the longer runtime. Its slow progress is likely due to numerical instabilities in the Hessian computation which do not occur in first-order methods.

## Acknowledgements

Research reported in this paper was partially supported through the Research Campus Modal funded by the German Federal Ministry of Education and Research (fund numbers 05M14ZAM,05M20ZBM) and the Deutsche Forschungsgemeinschaft (DFG) through the DFG Cluster of Excellence MATH+. We would like to thank the anonymous reviewers for their suggestions and comments.

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
