## Supplementary material

**Outline.**  The appendix of the paper is organized as follows:

- Section A presents the full convergence proof of the Frank-Wolfe gap for the Monotonous Frank-Wolfe algorithm, and improved convergence bounds when using the Frank-Wolfe algorithm with the step size strategy of Pedregosa et al. (2020) when the optimum is contained in the interior of $\mathcal{X} \cap \text{dom}(f)$, or when the feasible region is uniformly convex.
- Section B reviews the Away-step Frank-Wolfe algorithm (Guélat & Marcotte, 1986; Lacoste-Julien & Jaggi, 2015), and shows how using the step size strategy of Pedregosa et al. (2020) one can show a linear convergence in primal gap and in Frank-Wolfe gap when the feasible region is a polytope.
- Section C presents additional information about the experimental section of the paper.

## A   Monotonous Frank-Wolfe

This appendix contains the theoretical proofs that have not been included in the main body of the paper due to space constraints, as well as several remarks of interest. We start off with a remark regarding the convergence proof in Theorem 2.5, and continue by showing that the Monotonous Frank-Wolfe algorithm (restated for convenience in Algorithm 3) not only contracts the primal gap at a rate of $O(1/t)$ where $t$ is the iteration count, but also ensures that the minimum of the Frank-Wolfe gap over the run of the algorithm is bounded by $O(1/t)$ from above.

---

**Algorithm 3** Monotonous Frank-Wolfe (`M-FW`)

---

**Input:**  Point $\mathbf{x}_0 \in \mathcal{X} \cap \text{dom}(f)$, function $f$
**Output:**  Iterates $\mathbf{x}_1, \ldots \in \mathcal{X}$

---

1: **for** $t = 0$ **to** $\ldots$ **do**
2:     $\mathbf{v}_t \leftarrow \text{argmin}_{\mathbf{v} \in \mathcal{X}} \langle \nabla f(\mathbf{x}_t), \mathbf{v} \rangle$
3:     $\gamma_t \leftarrow 2/(t+2)$
4:     $\mathbf{x}_{t+1} \leftarrow \mathbf{x}_t + \gamma_t(\mathbf{v}_t - \mathbf{x}_t)$
5:     **if** $\mathbf{x}_{t+1} \notin \text{dom}(f)$  **or**  $f(\mathbf{x}_{t+1}) > f(\mathbf{x}_t)$ **then**
6:         $\mathbf{x}_{t+1} \leftarrow \mathbf{x}_t$

---

**Remark A.1** (Regarding the proof of Theorem 2.5)**.**  One of the quantities that we have used in the proof of Theorem 2.5 is $L_f^{\mathcal{L}_0}$. Note that the function $f$ is $L_f^{\mathcal{L}_0}$-smooth over $\mathcal{L}_0$. One could wonder then why we have bothered to use the bounds on the Bregman divergence in Proposition 2.2 for a $(M, \nu)$-generalized self-concordant function, instead of simply using the bounds from the $L_f^{\mathcal{L}_0}$-smoothness of $f$ over $\mathcal{L}_0$. The reason is that the upper bound on the Bregman divergence in Proposition 2.2 applies for any $\mathbf{x}, \mathbf{y} \in \text{dom}(f)$ such that $d_\nu(\mathbf{x}, \mathbf{y}) < 1$, and we can easily bound the number of iterations $T_\nu$ it takes for the step size $\gamma_t = 2/(t+2)$ to verify both $\mathbf{x}_t, \mathbf{x}_t + \gamma_t(\mathbf{v}_t - \mathbf{x}_t) \in \text{dom}(f)$ and $d_\nu(\mathbf{x}_t, \mathbf{x}_t + \gamma_t(\mathbf{v}_t - \mathbf{x}_t)) < 1$ for $t \geq T_\nu$. However, in order to apply the bounds on the Bregman divergence we need $\mathbf{x}_t, \mathbf{x}_t + \gamma_t(\mathbf{v}_t - \mathbf{x}_t) \in \mathcal{L}_0$, and while it is easy to show by monotonicity that $\mathbf{x}_t \in \mathcal{L}_0$, there is no straightforward way to prove that for some $\tilde{T}_\nu$ we have that $\mathbf{x}_t + \gamma_t(\mathbf{v}_t - \mathbf{x}_t) \in \mathcal{L}_0$ for all $t \geq \tilde{T}_\nu$.

As mentioned in Remark 2.1 in the main body of the text, in practice, a halving strategy for the step size is preferred for the implementation of the Monotonous Frank-Wolfe algorithm, as opposed to the step size implementation shown in Algorithm 3 (or Algorithm 1 in the main body).  This halving strategy, which is shown in Algorithm 4, helps deal with the

case in which a large number of **consecutive** step sizes $\gamma_t$ are rejected either because $\mathbf{x}_t + \gamma_t(\mathbf{v}_t - \mathbf{x}_t) \notin \mathrm{dom}(f)$ or $f(\mathbf{x}_t + \gamma_t(\mathbf{v}_t - \mathbf{x}_t)) > f(\mathbf{x}_t)$, and helps avoid the need to potentially call the zeroth-order or domain oracle a large number of times in these cases. The halving strategy in Algorithm 4 results in a step size that is at most a factor of 2 smaller than the one that would have been accepted with the original strategy, i.e., that would have ensured that $\mathbf{x}_t + \gamma_t(\mathbf{v}_t - \mathbf{x}_t) \in \mathrm{dom}(f)$ and $f(\mathbf{x}_t + \gamma_t(\mathbf{v}_t - \mathbf{x}_t)) \leq f(\mathbf{x}_t)$, in the standard Monotonous Frank-Wolfe algorithm in Algorithm 3. However, the number of zeroth-order or domain oracles that would be needed to find this step size that satisfies both $\mathbf{x}_t + \gamma_t(\mathbf{v}_t - \mathbf{x}_t) \in \mathrm{dom}(f)$ and $f(\mathbf{x}_{t+1}) \leq f(\mathbf{x}_t)$ is logarithmic for the Monotonous Frank-Wolfe variant shown in Algorithm 4, when compared to the number needed for the Monotonous Frank-Wolfe variant with halving shown in Algorithm 3. Note that the convergence properties established throughout the paper for the Monotonous Frank-Wolfe algorithm in Algorithm 3 also hold for the variant in Algorithm 4; with the only difference being that we lose a very small constant factor (e.g., at most a factor of 2 for the standard case) in the convergence rate.

---

**Algorithm 4** Halving Monotonous Frank-Wolfe

---

**Input:** Point $\mathbf{x}_0 \in \mathcal{X} \cap \mathrm{dom}(f)$, function $f$
**Output:** Iterates $\mathbf{x}_1, \ldots \in \mathcal{X}$

---

1: $\psi_{-1} \leftarrow 0$
2: **for** $t = 0$ **to** $\ldots$ **do**
3: $\quad \mathbf{v}_t \leftarrow \mathrm{argmin}_{\mathbf{v} \in \mathcal{X}} \langle \nabla f(\mathbf{x}_t), \mathbf{v} \rangle$
4: $\quad \psi_t \leftarrow \psi_{t-1}$
5: $\quad \gamma_t \leftarrow 2^{1-\psi_t}/(t+2)$
6: $\quad \mathbf{x}_{t+1} \leftarrow \mathbf{x}_t + \gamma_t(\mathbf{v}_t - \mathbf{x}_t)$
7: $\quad$ **while** $\mathbf{x}_{t+1} \notin \mathrm{dom}(f)$ **or** $f(\mathbf{x}_{t+1}) > f(\mathbf{x}_t)$ **do**
8: $\quad\quad \psi_t \leftarrow \psi_t + 1$
9: $\quad\quad \psi_t \leftarrow 2^{1-\psi_t}/(t+2)$
10: $\quad\quad \mathbf{x}_{t+1} \leftarrow \mathbf{x}_t + \gamma_t(\mathbf{v}_t - \mathbf{x}_t)$

---

## A.1 Convergence of the Frank-Wolfe gap for the Monotonous Frank-Wolfe algorithm

In this section of the appendix, we show that when running the Monotonous Frank-Wolfe algorithm (Algorithm 1) the minimum of the Frank-Wolfe gap over the run of the algorithm converges at a rate of $O(1/t)$. The idea of the proof is very similar to the one in Jaggi (2013). In a nutshell, as the primal progress per iteration is directly related to the step size times the Frank-Wolfe gap, we know that the Frank-Wolfe gap cannot remain indefinitely above a given value, as otherwise we would obtain a large amount of primal progress, which would make the primal gap become negative. This is formalized in Theorem A.2.

**Theorem A.2.** *Suppose $\mathcal{X}$ is a compact convex set and $f$ is a $(M, \nu)$ generalized self-concordant function with $\nu \geq 2$. Then if the Monotonous Frank-Wolfe algorithm (Algorithm 1) is run for $T \geq T_\nu + 6$ iterations, we will have that:*

$$\min_{1 \leq t \leq T} g(\mathbf{x}_t) \leq O(1/T),$$

*where $T_\nu$ is defined as:*

$$T_\nu \overset{\text{def}}{=} \begin{cases} \lceil 4MD \rceil - 2 & \text{if } \nu = 2 \\ \left\lceil 2MD(L_f^{\mathcal{L}_0})^{\nu/2-1}(\nu - 2) \right\rceil - 2 & \text{otherwise.} \end{cases} \tag{A.1}$$

*Proof.* In order to prove the claim, we focus on the iterations $T_\nu + \lceil (T - T_\nu)/3 \rceil - 2 \leq t \leq T - 2$, where $T_\nu$ is defined in Equation (A.1). Note that as we assume that $T \geq T_\nu + 6$, we know that $T_\nu \leq T_\nu + \lceil (T - T_\nu)/3 \rceil - 2$, and so for iterations $T_\nu + \lceil (T - T_\nu)/3 \rceil - 2 \leq t \leq T - 2$ we know that $d_\nu(\mathbf{x}_t, \mathbf{x}_{t+1}) \leq 1/2$, and so:

$$h(\mathbf{x}_{t+1}) \leq h(\mathbf{x}_t) - \gamma_t g(\mathbf{x}_t) + \gamma_t^2 L_f^{\mathcal{L}_0} D^2 \omega_\nu(1/2). \tag{A.2}$$

In a very similar fashion as was done in the proof of Theorem 2.5, we divide the proof into two different cases.

**Case** $-\gamma_t g(\mathbf{x}_t) + \gamma_t^2 L_f^{\mathcal{L}_0} D^2 \omega_\nu(1/2) \geq 0$ **for some** $T_\nu + \lceil (T - T_\nu)/3 \rceil - 2 \leq t \leq T - 2$**:** Reordering the inequality above we therefore know that there exists a $T_\nu + \lceil (T - T_\nu)/3 \rceil - 2 \leq K \leq T - 2$ such that:

$$g(\mathbf{x}_K) \leq \frac{2}{2 + K} L_f^{\mathcal{L}_0} D^2 \omega_\nu(1/2)$$

$$\leq \frac{2}{T_\nu + \lceil (T - T_\nu)/3 \rceil} L_f^{\mathcal{L}_0} D^2 \omega_\nu(1/2)$$

$$= \frac{6}{2T_\nu + T} L_f^{\mathcal{L}_0} D^2 \omega_\nu(1/2),$$

where the second inequality follows from the fact that $T_\nu + \lceil (T - T_\nu)/3 \rceil - 2 \leq K$. This leads to $\min_{1 \leq t \leq T} g(\mathbf{x}_t) \leq g(\mathbf{x}_K) \leq \frac{6}{2T_\nu + T} L_f^{\mathcal{L}_0} D^2 \omega_\nu(1/2)$.

**Case** $-\gamma_t g(\mathbf{x}_t) + \gamma_t^2 L_f^{\mathcal{L}_0} D^2 \omega_\nu(1/2) < 0$ **for all** $T_\nu + T_\nu + \lceil (T - T_\nu)/3 \rceil - 2 \leq t \leq T - 2$**:** Using the inequality above and plugging into Equation (A.2) allows us to conclude that all steps $T_\nu + T_\nu + \lceil (T - T_\nu)/3 \rceil - 2 \leq t \leq T - 2$ will produce primal progress using the step size $\gamma_t$, and so as we know that $\mathbf{x}_{t+1} \in \text{dom}(f)$ by Lemma 2.4, then for all $T_\nu + \lceil (T - T_\nu)/3 \rceil - 2 \leq t \leq T - 2$ we will take a non-zero step size determined by $\gamma_t$, as $\mathbf{x}_t + \gamma_t(\mathbf{v}_t - \mathbf{x}_t) \in \text{dom}(f)$ and $f(\mathbf{x}_t + \gamma_t(\mathbf{v}_t - \mathbf{x}_t)) < f(\mathbf{x}_t)$ in Line 5 of Algorithm 1. Consequently, summing up Equation (A.2) from $t_{\min} \stackrel{\text{def}}{=} T_\nu + \lceil (T - T_\nu)/3 \rceil - 2$ to $t_{\max} \stackrel{\text{def}}{=} T - 2$ we have that:

$$h(\mathbf{x}_{t_{\max}+1}) \leq h\left(\mathbf{x}_{t_{\min}}\right) - \sum_{t=t_{\min}}^{t_{\max}} \gamma_t g(\mathbf{x}_t) + L_f^{\mathcal{L}_0} D^2 \omega_\nu(1/2) \sum_{t=t_{\min}}^{t_{\max}} \gamma_t^2 \tag{A.3}$$

$$\leq h\left(\mathbf{x}_{t_{\min}}\right) - 2 \min_{t_{\min} \leq t \leq t_{\max}} g(\mathbf{x}_t) \sum_{t=t_{\min}}^{t_{\max}} \frac{1}{2 + t} + 4 L_f^{\mathcal{L}_0} D^2 \omega_\nu(1/2) \sum_{t=t_{\min}}^{t_{\max}} \frac{1}{(2 + t)^2} \tag{A.4}$$

$$\leq h\left(\mathbf{x}_{t_{\min}}\right) - 2 \min_{1 \leq t \leq T} g(\mathbf{x}_t) \frac{t_{\max} - t_{\min} + 1}{2 + t_{\max}} + 4 L_f^{\mathcal{L}_0} D^2 \omega_\nu(1/2) \frac{t_{\max} - t_{\min} + 1}{(2 + t_{\min})^2} \tag{A.5}$$

$$\leq 4 \left( \frac{T_\nu + 1}{t_{\min} + 1} + \frac{t_{\max} - t_{\min} + 1}{(2 + t_{\min})^2} \right) \max \left\{ h(\mathbf{x}_0), L_f^{\mathcal{L}_0} D^2 \omega_\nu(1/2) \right\} \tag{A.6}$$

$$- 2 \min_{1 \leq t \leq T} g(\mathbf{x}_t) \frac{t_{\max} - t_{\min} + 1}{2 + t_{\max}}. \tag{A.7}$$

Note that Equation A.4 stems from the fact that $\min_{t_{\min} \leq t \leq t_{\max}} g(\mathbf{x}_t) \leq g(\mathbf{x}_t)$ for any $t_{\min} \leq t \leq t_{\max}$, and from plugging $\gamma_t = 2/(2 + t)$, and Equation A.5 follows from the fact that $-1/(2 + t) \leq -1/(2 + t_{\max})$ and $1/(2 + t) \leq 1/(2 + t_{\min})$ for all $t_{\min} \leq t \leq t_{\max}$. The last inequality, Equation (A.6) and (A.7) arises from plugging in the upper bound on the primal gap $h(\mathbf{x}_{t_{\min}})$ from Theorem 2.5 and collecting terms. If we plug in the specific values of $t_{\max}$ and $t_{\min}$ this leads to:

$$h(\mathbf{x}_{T-2}) \leq 12 \left( \frac{T_\nu + 1}{2T_\nu + T - 3} + \frac{2T - 2T_\nu + 3}{(2T_\nu + T)^2} \right) \max \left\{ h(\mathbf{x}_0), L_f^{\mathcal{L}_0} D^2 \omega_\nu(1/2) \right\} \tag{A.8}$$

$$- \frac{2}{3} \min_{1 \leq t \leq T} g(\mathbf{x}_t) \frac{T - T_\nu}{T}. \tag{A.9}$$

We establish our claim using proof by contradiction. Assume that:

$$\min_{1 \leq t \leq T} g(\mathbf{x}_t) > \frac{18T}{T - T_\nu} \left( \frac{T_\nu + 1}{2T_\nu + T - 3} + \frac{2T - 2T_\nu + 3}{(2T_\nu + T)^2} \right) \max \left\{ h(\mathbf{x}_0), L_f^{\mathcal{L}_0} D^2 \omega_\nu(1/2) \right\}.$$

Then by plugging into the bound in Equation (A.9) we have that $h(\mathbf{x}_{T-2}) < 0$, which is the desired contradiction, as the primal gap cannot be negative. Therefore we must have that:

$$\min_{1 \leq i \leq T} g(\mathbf{x}_i) \leq \frac{18T}{T - T_\nu} \left( \frac{T_\nu + 1}{2T_\nu + T - 3} + \frac{2T - 2T_\nu + 3}{(2T_\nu + T)^2} \right) \max \left\{ h(\mathbf{x}_0), L_f^{\mathcal{L}_0} D^2 \omega_\nu(1/2) \right\} = \mathcal{O}(1/T).$$

This completes the proof. $\qquad\square$

## A.2 Improved convergence bounds

We focus on two different settings to obtain improved convergence rates; in the first, we assume that $\mathbf{x}^* \in \text{Int}\,(\mathcal{X} \cap \text{dom}(f))$ (Section A.2), and in the second we assume that $\mathcal{X}$ is strongly or uniformly convex (Section A.2). In this section we focus on the combination of a slightly modified Frank-Wolfe algorithm with the adaptive line search technique of Pedregosa et al. (2020) (shown for reference in Algorithm 5 and 6). This is the same algorithm used in Dvurechensky et al. (2020b), however, we show improved convergence rates in several settings of interest.

---

**Algorithm 5** Monotonous Frank-Wolfe with the adaptive step size of Pedregosa et al. (2020)

---

**Input:** Point $\mathbf{x}_0 \in \mathcal{X} \cap \text{dom}(f)$, function $f$, initial smoothness estimate $L_{-1}$
**Output:** Iterates $\mathbf{x}_1, \ldots \in \mathcal{X}$

---

1: **for** $t = 0$ **to** $\ldots$ **do**
2: $\quad \mathbf{v}_t \leftarrow \text{argmin}_{\mathbf{v} \in \mathcal{X}} \langle \nabla f(\mathbf{x}_t), \mathbf{v} \rangle$
3: $\quad \gamma_t, L_t \leftarrow \texttt{Backtrack}(f, \mathbf{x}_t, \mathbf{v}_t - \mathbf{x}_t, L_{t-1}, 1)$
4: $\quad \mathbf{x}_{t+1} \leftarrow \mathbf{x}_t + \gamma_t(\mathbf{v}_t - \mathbf{x}_t)$

---

**Algorithm 6** $\texttt{Backtrack}(f, \mathbf{x}, \mathbf{d}, L_{t-1}, \gamma_{\max})$ (line search of Pedregosa et al. (2020))

---

**Input:** Point $\mathbf{x} \in \mathcal{X} \cap \text{dom}(f)$, $\mathbf{v} \in \mathbb{R}^n$, function $f$, estimate $L_{t-1}$, step size $\gamma_{\max}$
**Output:** $\gamma, M$

---

1: Choose $\tau > 1$, $\eta \leq 1$ and $M \in [\eta L_{t-1}, L_{t-1}]$
2: $\gamma = \min\{-\langle \nabla f(\mathbf{x}), \mathbf{d} \rangle / (M \|\mathbf{d}\|^2), \gamma_{\max}\}$
3: **while** $\mathbf{x} + \gamma \mathbf{d} \notin \text{dom}(f)$ **or** $f(\mathbf{x} + \gamma \mathbf{d}) - f(\mathbf{x}) > \frac{M\gamma^2}{2} \|\mathbf{d}\|^2 + \gamma \langle \nabla f(\mathbf{x}), \mathbf{d} \rangle$ **do**
4: $\quad M = \tau M$
5: $\quad \gamma = \min\{-\langle \nabla f(\mathbf{x}), \mathbf{d} \rangle / (M \|\mathbf{d}\|^2), \gamma_{\max}\}$

---

### Optimum contained in the interior

Before proving the main theoretical results of this section we first review some auxiliary results that allow us to prove the linear convergence in this setting.

**Theorem A.3.** *(C.f., (Nesterov, 2018, Theorem 5.1.6)) Let $f$ be generalized self-concordant and $\text{dom}(f)$ not contain straight lines, then the Hessian $\nabla^2 f(\mathbf{x})$ is non-degenerate at all points $\mathbf{x} \in \text{dom}(f)$.*

Note that the assumption that $\text{dom}(f)$ does not contain straight lines is without loss of generality as we can simply modify the function outside of our compact convex feasible region so that it holds.

**Proposition A.4.** *(C.f., Guélat & Marcotte (1986)) If there exists an $r > 0$ such that $\mathcal{B}(\mathbf{x}^*, r) \subseteq \mathcal{X} \cap \text{dom}(f)$, then for all $\mathbf{x} \in \mathcal{X} \cap \text{dom}(f)$ we have that:*

$$\frac{g(\mathbf{x})}{\|\mathbf{x} - \mathbf{v}\|} \geq \frac{r}{D} \|\nabla f(\mathbf{x})\| \geq \frac{r}{D} \frac{\langle \nabla f(\mathbf{x}), \mathbf{x} - \mathbf{x}^* \rangle}{\|\mathbf{x} - \mathbf{x}^*\|},$$

*where $\mathbf{v} = \underset{\mathbf{y} \in \mathcal{X}}{\text{argmin}} \langle \nabla f(\mathbf{x}), \mathbf{y} \rangle$ and $g(\mathbf{x})$ is the Frank-Wolfe gap.*

With these tools at hand, we have that the Frank-Wolfe algorithm with the backtracking step size strategy converges at a linear rate.

**Theorem A.5.** *Let $f$ be a $(M, \nu)$ generalized self-concordant function with $\nu \geq 2$ and let $\text{dom}(f)$ not contain straight lines. Furthermore, we denote by $r > 0$ the largest value such that $\mathcal{B}(\mathbf{x}^*, r) \subseteq \mathcal{X} \cap \text{dom}(f)$. Then the Frank-Wolfe algorithm (Algorithm 5) with the backtracking strategy of Pedregosa et al. (2020) results in a convergence:*

$$h(\mathbf{x}_t) \leq h(\mathbf{x}_0) \left( 1 - \frac{\mu_f^{\mathcal{L}_0}}{2\tilde{L}} \left( \frac{r}{D} \right)^2 \right)^t,$$

*for $t \geq 1$, where $\tilde{L} \overset{\text{def}}{=} \max\{\tau L_f^{\mathcal{L}_0}, L_{-1}\}$, $\tau > 1$ is the backtracking parameter, $L_{-1}$ is the initial smoothness estimate in Algorithm 6, $\mu_f^{\mathcal{L}_0} = \min\limits_{\mathbf{u} \in \mathcal{L}_0, \mathbf{d} \in \mathbb{R}^n} \|\mathbf{d}\|^2_{\nabla^2 f(\mathbf{u})} / \|\mathbf{d}\|^2_2$ and $L_f^{\mathcal{L}_0} = \max\limits_{\mathbf{u} \in \mathcal{L}_0, \mathbf{d} \in \mathbb{R}^n} \|\mathbf{d}\|^2_{\nabla^2 f(\mathbf{u})} / \|\mathbf{d}\|^2_2$.*

*Proof.* Consider the compact set $\mathcal{L}_0 \overset{\text{def}}{=} \{\mathbf{x} \in \text{dom}(f) \cap \mathcal{X} \mid f(\mathbf{x}) \leq f(\mathbf{x}_0)\}$. As the backtracking line search makes monotonous primal progress, we know that for $t \geq 0$ we will have that $\mathbf{x}_t \in \mathcal{L}_0$. Consequently we can define $\mu_f^{\mathcal{L}_0} = \min_{\mathbf{u} \in \mathcal{L}_0, \mathbf{d} \in \mathbb{R}^n} \|\mathbf{d}\|^2_{\nabla^2 f(\mathbf{u})} / \|\mathbf{d}\|^2_2$. As by our assumption $\text{dom}(f)$ does not contain any straight lines we know that for all $\mathbf{x} \in \text{dom}(f)$ the Hessian is non-degenerate, and therefore $\mu_f^{\mathcal{L}_0} > 0$. This allows us to claim that for any $\mathbf{x}, \mathbf{y} \in \mathcal{L}_0$ we have that:

$$f(\mathbf{x}) - f(\mathbf{y}) - \langle \nabla f(\mathbf{y}), \mathbf{x} - \mathbf{y} \rangle \geq \frac{\mu_f^{\mathcal{L}_0}}{2} \|\mathbf{x} - \mathbf{y}\|^2. \tag{A.10}$$

The backtracking line search in Algorithm 6 will either output a point $\gamma_t = 1$ or $\gamma_t < 1$. In any case, Algorithm 6 will find and output a smoothness estimate $L_t$ and a step size $\gamma_t$ such that for $\mathbf{x}_{t+1} = \mathbf{x}_t + \gamma_t (\mathbf{v}_t - \mathbf{x}_t)$ we have that:

$$f(\mathbf{x}_{t+1}) - f(\mathbf{x}_t) \leq \frac{L_t \gamma_t^2}{2} \|\mathbf{x}_t - \mathbf{v}_t\|^2 - \gamma_t g(\mathbf{x}_t). \tag{A.11}$$

In the case where $\gamma_t = 1$ we know by observing Line 5 of Algorithm 6 that $g(\mathbf{x}_t) \geq L_t \|\mathbf{x}_t - \mathbf{v}_t\|^2$, and so plugging into Equation (A.11) we arrive at $h(\mathbf{x}_{t+1}) \leq h(\mathbf{x}_t)/2$. In the case where $\gamma_t = g(\mathbf{x}_t)/(L_t \|\mathbf{x}_t - \mathbf{v}_t\|^2) < 1$, we have that $g(\mathbf{x}_t) < L_t \|\mathbf{x}_t - \mathbf{v}_t\|^2$, which leads to $h(\mathbf{x}_{t+1}) \leq h(\mathbf{x}_k) - g(\mathbf{x}_t)^2/(2L_t \|\mathbf{x}_t - \mathbf{v}_t\|^2)$, when plugging the expression for the step size in the progress bound in Equation A.11. In this last case where $\gamma_t < 1$ we have the following contraction for the primal gap:

$$\begin{aligned} h(\mathbf{x}_t) - h(\mathbf{x}_{t+1}) &\geq \frac{g(\mathbf{x}_t)^2}{2L_t \|\mathbf{x}_t - \mathbf{v}_t\|^2} \\ &\geq \frac{r^2}{D^2} \frac{\|\nabla f(\mathbf{x}_t)\|^2}{2L_t} \\ &\geq \frac{\mu_f^{\mathcal{L}_0}}{\tilde{L}} \frac{r^2}{D^2} h(\mathbf{x}_t), \end{aligned}$$

where we have used the inequality that involves the central term and the leftmost term in Proposition A.4, and the last inequality stems from the bound $h(\mathbf{x}_t) \leq \|\nabla f(\mathbf{x}_t)\|^2 / (2\mu_f^{\mathcal{L}_0})$ for $\mu_f^{\mathcal{L}_0}$-strongly convex functions. Putting the above bounds together we have that:

$$h(\mathbf{x}_{t+1}) \leq h(\mathbf{x}_t)\left(1 - \frac{1}{2} \min\left\{1, \frac{\mu_f^{\mathcal{L}_0}}{\tilde{L}}\left(\frac{r}{D}\right)^2\right\}\right)$$

$$\leq h(\mathbf{x}_t)\left(1 - \frac{\mu_f^{\mathcal{L}_0}}{2\tilde{L}}\left(\frac{r}{D}\right)^2\right),$$

which completes the proof. $\qquad\square$

The previous bound depends on the largest positive $r$ such that $\mathcal{B}(\mathbf{x}^*, r) \subseteq \mathcal{X} \cap \text{dom}(f)$, which can be arbitrarily small. Note also that the previous proof uses the lower bound of the Bregman divergence from the $\mu_f^{\mathcal{L}_0}$-strong convexity of the function over $\mathcal{L}_0$ to obtain linear convergence. Note that this bound is local, and is only of use because the step size strategy of Algorithm 6 automatically ensures that if $\mathbf{x}_t \in \mathcal{L}_0$ and $\mathbf{d}_t$ is a direction of descent, then $\mathbf{x}_t + \gamma_t \mathbf{d}_t \in \mathcal{L}_0$. This is in contrast with Algorithm 3, in which the step size $\gamma_t = 2/(2+t)$ did

not automatically ensure monotonicity in primal gap, and this had to be enforced by setting $\mathbf{x}_{t+1} = \mathbf{x}_t$ if $f(\mathbf{x}_t + \gamma_t \mathbf{d}_t) > f(\mathbf{x}_t)$, where $\mathbf{d}_t = \mathbf{v}_t - \mathbf{x}_t$. If we were to have used the lower bound on the Bregman divergence from Sun & Tran-Dinh (2019, Proposition 10) in the proof, which states that:

$$f(\mathbf{y}) - f(\mathbf{x}) - \langle \nabla f(\mathbf{x}), \mathbf{y} - \mathbf{x} \rangle \geq \omega_\nu(-d_\nu(\mathbf{x} - \mathbf{y})) \|\mathbf{y} - \mathbf{x}\|^2_{\nabla^2 f(\mathbf{x})},$$

for any $\mathbf{x}, \mathbf{y} \in \text{dom}(f)$ and any $\nu \geq 2$, we would have arrived at a bound that holds over all $\text{dom}(f)$. However, in order to arrive at a usable bound, and armed only with the knowledge that the Hessian is non-degenerate if $\text{dom}(f)$ does not contain straight lines, and that $\mathbf{x}, \mathbf{y} \in \mathcal{L}_0$, we would have had to write:

$$\omega_\nu(-d_\nu(\mathbf{x} - \mathbf{y})) \|\mathbf{x} - \mathbf{y}\|^2_{\nabla^2 f(\mathbf{y})} \geq \mu_f^{\mathcal{L}_0} \omega_\nu(-d_\nu(\mathbf{x} - \mathbf{y})) \|\mathbf{x} - \mathbf{y}\|^2,$$

where the inequality follows from the definition of $\mu_f^{\mathcal{L}_0}$. It is easy to see that as $d\omega_\nu(\tau)/d\tau > 0$ by Remark 2.3, we have that $1/2 = \omega_\nu(0) \geq \omega_\nu(-d_\nu(\mathbf{x} - \mathbf{y}))$. This results in a bound:

$$f(\mathbf{y}) - f(\mathbf{x}) - \langle \nabla f(\mathbf{x}), \mathbf{y} - \mathbf{x} \rangle \geq \mu_f^{\mathcal{L}_0} \omega_\nu(-d_\nu(\mathbf{x} - \mathbf{y})) \|\mathbf{x} - \mathbf{y}\|^2. \tag{A.12}$$

When we compare the bounds on Equation (A.10) and (A.12), we can see that the bound from $\mu_f^{\mathcal{L}_0}$-strong convexity is tighter than the bound from the properties of $(M, \nu)$-generalized self-concordant functions, albeit local. This is the reason why we have used the former bound in the proof of Theorem A.5.

**Strongly convex or uniformly convex sets**

In order to prove convergence rate results for the case where the feasible region is $(\kappa, p)$-uniformly convex, we first review the definition of the $(\kappa, p)$-uniform convexity of a set (see Definition A.6), as well as a useful lemma that allows us to go from contractions to convergence rates.

**Definition A.6** ($(\kappa, q)$-uniformly convex set). Given two positive numbers $\kappa$ and $q$, we say the set $\mathcal{X} \subseteq \mathbb{R}^n$ is $(\kappa, q)$-*uniformly convex* with respect to a norm $\|\cdot\|$ if for any $\mathbf{x}, \mathbf{y} \in \mathcal{X}$, $0 \leq \gamma \leq 1$, and $\mathbf{z} \in \mathbb{R}^n$ with $\|\mathbf{z}\| = 1$ we have that:

$$\mathbf{y} + \gamma(\mathbf{x} - \mathbf{y}) + \gamma(1 - \gamma) \cdot \kappa \|\mathbf{x} - \mathbf{y}\|^q \mathbf{z} \in \mathcal{X}.$$

The previous definition allows us to obtain a scaling inequality very similar to the one shown in Theorem A.4, which is key to proving the following convergence rates, and can be implicitly found in Kerdreux et al. (2021) and Garber & Hazan (2016).

**Proposition A.7.** *Let* $\mathcal{X} \subseteq \mathbb{R}^n$ *be* $(\kappa, q)$-uniformly convex, *then for all* $\mathbf{x} \in \mathcal{X}$:

$$\frac{g(\mathbf{x})}{\|\mathbf{x} - \mathbf{v}\|^q} \geq \kappa \|\nabla f(\mathbf{x})\|,$$

*where* $\mathbf{v} = \text{argmin}_{\mathbf{u} \in \mathcal{X}} \langle \nabla f(\mathbf{x}), \mathbf{u} \rangle$, *and* $g(\mathbf{x})$ *is the Frank-Wolfe gap.*

The next lemma that will be presented is an extension of the one used in Kerdreux et al. (2021, Lemma A.1) (see also Temlyakov (2015)), and allows us to go from per iteration contractions to convergence rates.

**Lemma A.8.** *We denote a sequence of nonnegative numbers by* $\{h_t\}_t$. *Let* $c_0$, $c_1$, $c_2$ *and* $\alpha$ *be positive numbers such that* $c_1 < 1$, $h_1 \leq c_0$ *and* $h_t - h_{t+1} \geq h_t \min\{c_1, c_2 h_t^\alpha\}$ *for* $t \geq 1$, *then:*

$$h_t \leq \begin{cases} c_0 (1 - c_1)^{t-1} & \text{if } 1 \leq t \leq t_0 \\ \frac{(c_1/c_2)^{1/\alpha}}{(1 + c_1 \alpha(t - t_0))^{1/\alpha}} = O\left(t^{-1/\alpha}\right) & \text{otherwise.} \end{cases}$$

*where*

$$t_0 \stackrel{\text{def}}{=} \max \left\{ 1, \left\lfloor \log_{1-c_1}\left(\frac{(c_1/c_2)^{1/\alpha}}{c_0}\right) \right\rfloor \right\}.$$

This allows us to conveniently transform the per iteration contractions to convergence rates. Moving on to the proof of the convergence rate.

**Theorem A.9.** *Suppose $X$ is a compact $(\kappa, q)$-uniformly convex set and $f$ is a $(M, \nu)$ generalized self-concordant function with $\nu \geq 2$. Furthermore, assume that $\min_{\mathbf{x} \in X} \|\nabla f(\mathbf{x})\| \geq C$. Then the Frank-Wolfe algorithm with Backtrack (Algorithm 5) results in a convergence:*

$$h_t \leq \begin{cases} h(\mathbf{x}_0) \left(1 - \frac{1}{2}\min\left\{1, \frac{\kappa C}{\tilde{L}}\right\}\right)^t & \text{if } q = 2 \\ \frac{h(\mathbf{x}_0)}{2^t} & \text{if } q > 2, 1 \leq t \leq t_0 \\ \frac{(\tilde{L}^q/(\kappa C)^2)^{1/(q-2)}}{(1+(q-2)(t-t_0)/(2q))^{q/(q-2)}} = O\left(t^{-q/(q-2)}\right) & \text{if } q > 2, t > t_0, \end{cases}$$

*for $t \geq 1$, where:*

$$t_0 = \max\left\{1, \left\lfloor \log_{1/2}\left(\frac{(\tilde{L}^q/(\kappa C)^2)^{1/(q-2)}}{h(\mathbf{x}_0)}\right) \right\rfloor\right\}.$$

*and $\tilde{L} \stackrel{\text{def}}{=} \max\{\tau L_f^{\mathcal{L}_0}, L_{-1}\}$, where $\tau > 1$ is the backtracking parameter, $L_{-1}$ is the initial smoothness estimate in Algorithm 6, and $L_f^{\mathcal{L}_0} = \max_{\mathbf{u} \in \mathcal{L}_0, \mathbf{d} \in \mathbb{R}^n} \|\mathbf{d}\|_{\nabla^2 f(\mathbf{u})}^2 / \|\mathbf{d}\|_2^2$.*

*Proof.* At iteration $t$, the backtracking line search strategy finds through successive function evaluations a $L_t > 0$ such that:

$$h(\mathbf{x}_{t+1}) \leq h(\mathbf{x}_t) - \gamma_t g(\mathbf{x}_t) + \frac{L_t \gamma_t^2}{2}\|\mathbf{x}_t - \mathbf{v}_t\|^2.$$

Finding the $\gamma_t$ that maximizes the right-hand side of the previous inequality leads to $\gamma_t = \min\{1, g(\mathbf{x}_t)/(L_t\|\mathbf{x}_t - \mathbf{v}_t\|^2)\}$, which is the step size ultimately taken by the algorithm at iteration $t$. Note that if $\gamma_t = 1$ this means that $g(\mathbf{x}_t) \geq L_t\|\mathbf{x}_t - \mathbf{v}_t\|^2$, which when plugged into the inequality above leads to $h(\mathbf{x}_{t+1}) \leq h(\mathbf{x}_t)/2$. Conversely, for $\gamma_t < 1$ we have that $h(\mathbf{x}_{t+1}) \leq h(\mathbf{x}_t) - g(\mathbf{x}_t)^2/(2L_t\|\mathbf{x}_t - \mathbf{v}_t\|^2)$. Focusing on this case and using the bounds $g(\mathbf{x}_t) \geq h(\mathbf{x}_t)$ and $g(\mathbf{x}_t) \geq \kappa\|\nabla f(\mathbf{x}_t)\|\|\mathbf{x}_t - \mathbf{v}_t\|^q$ from Proposition A.7 leads to:

$$h(\mathbf{x}_{t+1}) \leq h(\mathbf{x}_t) - h(\mathbf{x}_t)^{2-2/q}\frac{(\kappa\|\nabla f(\mathbf{x}_t)\|)^{2/q}}{2L_t} \tag{A.13}$$

$$\leq h(\mathbf{x}_t) - h(\mathbf{x}_t)^{2-2/q}\frac{(\kappa C)^{2/q}}{2\tilde{L}}, \tag{A.14}$$

where the last inequality simply comes from the bound on the gradient norm, and the fact that $L_t \leq \tilde{L}$, for $\tilde{L} \stackrel{\text{def}}{=} \max\{\tau L_f^{\mathcal{L}_0}, L_{-1}\}$, where $\tau > 1$ is the backtracking parameter and $L_{-1}$ is the initial smoothness estimate in Algorithm 6. Reordering this expression and putting together the two cases we have that:

$$h(\mathbf{x}_t) - h(\mathbf{x}_{t+1}) \geq h(\mathbf{x}_t)\min\left\{\frac{1}{2}, \frac{(\kappa C)^{2/q}}{2\tilde{L}}h(\mathbf{x}_t)^{1-2/q}\right\}.$$

For the case where $q = 2$ we get a linear contraction in primal gap. Using Lemma A.8 to go from a contraction to a convergence rate for $q > 2$ we have that:

$$h_t \leq \begin{cases} h(\mathbf{x}_0)\left(1 - \frac{1}{2}\min\left\{1, \frac{\kappa C}{\tilde{L}}\right\}\right)^t & \text{if } q = 2 \\ \frac{h(\mathbf{x}_0)}{2^t} & \text{if } q > 2, 1 \leq t \leq t_0 \\ \frac{(\tilde{L}^q/(\kappa C)^2)^{1/(q-2)}}{(1+(q-2)(t-t_0)/(2q))^{q/(q-2)}} = O\left(t^{-q/(q-2)}\right) & \text{if } q > 2, t > t_0, \end{cases}$$

for $t \geq 1$, where:

$$t_0 = \max\left\{1, \left\lfloor \log_{1/2}\left(\frac{(\tilde{L}^q/(\kappa C)^2)^{1/(q-2)}}{h(\mathbf{x}_0)}\right) \right\rfloor\right\},$$

which completes the proof.
$\qquad\square$

Lastly, we deal with the general case in which the norm of the gradient is not bounded away from zero in $\mathcal{X}$.

**Theorem A.10.** *Suppose $\mathcal{X}$ is a compact $(\kappa, q)$-uniformly convex set and $f$ is a $(M, \nu)$ generalized self-concordant function with $\nu \geq 2$ for which domain does not contain straight lines. Then the Frank-Wolfe algorithm with Backtrack (Algorithm 5) results in a convergence:*

$$
h_t \leq \begin{cases} \dfrac{h(\mathbf{x}_0)}{2^t} & \text{if } 1 \leq t \leq t_0 \\ \dfrac{(\tilde{L}^q/(\kappa^2 \mu_f^{\mathcal{L}_0}))^{1/(q-1)}}{(1+(q-1)(t-t_0)/(2q))^{q/(q-1)}} = O\left(t^{-q/(q-1)}\right) & \text{if } t > t_0, \end{cases}
$$

*for $t \geq 1$, where:*

$$
t_0 = \max \left\{ 1, \left\lfloor \log_{1/2} \left( \frac{(\tilde{L}^q/(\kappa^2 \mu_f^{\mathcal{L}_0}))^{1/(q-1)}}{h(\mathbf{x}_0)} \right) \right\rfloor \right\}.
$$

*and $\tilde{L} \stackrel{\text{def}}{=} \max\{\tau L_f^{\mathcal{L}_0}, L_{-1}\}$, where $\tau > 1$ is the backtracking parameter, $L_{-1}$ is the initial smoothness estimate in Algorithm 6, $\mu_f^{\mathcal{L}_0} = \min_{\mathbf{u} \in \mathcal{L}_0, \mathbf{d} \in \mathbb{R}^n} \|\mathbf{d}\|_{\nabla^2 f(\mathbf{u})}^2 / \|\mathbf{d}\|_2^2$ and $L_f^{\mathcal{L}_0} = \max_{\mathbf{u} \in \mathcal{L}_0, \mathbf{d} \in \mathbb{R}^n} \|\mathbf{d}\|_{\nabla^2 f(\mathbf{u})}^2 / \|\mathbf{d}\|_2^2$.*

*Proof.* Consider the compact set $\mathcal{L}_0 \stackrel{\text{def}}{=} \{\mathbf{x} \in \text{dom}(f) \cap \mathcal{X} \mid f(\mathbf{x}) \leq f(\mathbf{x}_0)\}$. As the algorithm makes monotonous primal progress we have that $\mathbf{x}_t \in \mathcal{L}_0$ for $t \geq 0$. The proof proceeds very similarly as before, except for the fact that now we have to bound $\|\nabla f(\mathbf{x}_t)\|$ using $\mu_f^{\mathcal{L}_0}$-strong convexity for points $\mathbf{x}_t, \mathbf{x}_t + \gamma_t(\mathbf{v}_t - \mathbf{x}_t) \in \mathcal{L}_0$. Continuing from Equation (A.13) for the case where $\gamma_t < 1$ and using the fact that $h(\mathbf{x}_t) \leq \|\nabla f(\mathbf{x}_t)\|^2 / (2\mu_f^{\mathcal{L}_0})$ we have that:

$$
h(\mathbf{x}_{t+1}) \leq h(\mathbf{x}_t) - h(\mathbf{x}_t)^{2-2/q} \frac{(\kappa \|\nabla f(\mathbf{x}_t)\|)^{2/q}}{2L_t}
$$

$$
\leq h(\mathbf{x}_t) - h(\mathbf{x}_t)^{2-1/q} \frac{\kappa^{2/q}(\mu_f^{\mathcal{L}_0})^{1/q} 2^{1/q-1}}{\tilde{L}},
$$

where we have also used the bound $L_t \leq \tilde{L}$ in the last equation. This leads us to a contraction, together with the case where $\gamma_t = 1$, which is unchanged from the previous proofs, of the form:

$$
h(\mathbf{x}_t) - h(\mathbf{x}_{t+1}) \geq h(\mathbf{x}_t) \min \left\{ \frac{1}{2}, \frac{\kappa^{2/q}(\mu_f^{\mathcal{L}_0})^{1/q} 2^{1/q-1}}{\tilde{L}} h(\mathbf{x}_t)^{1-1/q} \right\}.
$$

Using again Lemma A.8 to go from a contraction to a convergence rate for $q > 2$ we have that:

$$
h_t \leq \begin{cases} \dfrac{h(\mathbf{x}_0)}{2^t} & \text{if } 1 \leq t \leq t_0 \\ \dfrac{(\tilde{L}^q/(\kappa^2 \mu_f^{\mathcal{L}_0}))^{1/(q-1)}}{(1+(q-1)(t-t_0)/(2q))^{q/(q-1)}} = O\left(t^{-q/(q-1)}\right) & \text{if } t > t_0, \end{cases}
$$

for $t \geq 1$, where:

$$
t_0 = \max \left\{ 1, \left\lfloor \log_{1/2} \left( \frac{(\tilde{L}^q/(\kappa^2 \mu_f^{\mathcal{L}_0}))^{1/(q-1)}}{h(\mathbf{x}_0)} \right) \right\rfloor \right\},
$$

which completes the proof. □

In Table 3 we provide an oracle complexity breakdown for the Frank-Wolfe algorithm with Backtrack (B-FW), also referred to as LBTFW-GSC in Dvurechensky et al. (2020b), when minimizing over a $(\kappa, q)$-uniformly convex set.

| Algorithm | Assumptions | FOO/ZOO/LMO/DO calls | Reference |
|---|---|---|---|
| B-FW/LBTFW-GSC$^\ddagger$ | $\mathbf{x}^* \in \mathrm{Int}\,(\mathcal{X} \cap \mathrm{dom}(f))$ | $\mathcal{O}(\log 1/\varepsilon)$ | **This work** |
| B-FW/LBTFW-GSC$^\ddagger$ | $\min_{\mathbf{x} \in \mathcal{X}} \|\nabla f(\mathbf{x})\| > 0, q = 2$ | $\mathcal{O}(\log 1/\varepsilon)$ | **This work** |
| B-FW/LBTFW-GSC$^\ddagger$ | $\min_{\mathbf{x} \in \mathcal{X}} \|\nabla f(\mathbf{x})\| > 0, q > 2$ | $\mathcal{O}\left(\varepsilon^{-(q-2)/q}\right)$ | **This work** |
| B-FW/LBTFW-GSC$^\ddagger$ | No straight lines in $\mathcal{X}$ | $\mathcal{O}\left(\varepsilon^{-(q-1)/q}\right)$ | **This work** |

Table 3: **Complexity comparison for** B-FW **(Algorithm 2) when minimizing over a** $(\kappa, q)$**-uniformly convex set**: Number of iterations needed to reach an $\varepsilon$-optimal solution in $h(\mathbf{x})$ for Problem 1.1 in several cases of interest. We use the superscript $\ddagger$ to indicate that constants in the convergence bounds depend on user-defined inputs.

## B   Away-step Frank-Wolfe

When the domain $\mathcal{X}$ is a polytope, one can obtain linear convergence in primal gap for a generalized self-concordant function using the well known *Away-step Frank-Wolfe* (AFW) algorithm (Guélat & Marcotte, 1986; Lacoste-Julien & Jaggi, 2015) with the adaptive step size of Pedregosa et al. (2020), shown in Algorithm 7. We use $\mathcal{S}_t$ to denote the *active set* at iteration $t$, that is, the set of vertices of the polytope that gives rise to $\mathbf{x}_t$ as a convex combination with positive weights. We can see that the algorithm either chooses to perform what is know as a *Frank-Wolfe* step in Line 6 of Algorithm 7 if the Frank-Wolfe gap $g(\mathbf{x})$ is greater than the *away gap* $\langle \nabla f(\mathbf{x}_t), \mathbf{a}_t - \mathbf{x}_t \rangle$ or an *Away*-step in Line 8 of Algorithm 7 otherwise.

---

**Algorithm 7** Away-step Frank-Wolfe (B-AFW) with the step size of Pedregosa et al. (2020)

---

**Input:** Point $\mathbf{x}_0 \in \mathcal{X} \cap \mathrm{dom}(f)$, function $f$, initial smoothness estimate $L_{-1}$
**Output:** Iterates $\mathbf{x}_1, \ldots \in \mathcal{X}$

---

1: $\mathcal{S}_0 \leftarrow \{\mathbf{x}_0\}$, $\boldsymbol{\lambda}_0 \leftarrow \{1\}$
2: **for** $t = 0$ **to** $\ldots$ **do**
3:      $\mathbf{v}_t \leftarrow \mathrm{argmin}_{\mathbf{v} \in \mathcal{X}} \langle \nabla f(\mathbf{x}_t), \mathbf{v} \rangle$
4:      $\mathbf{a}_t \leftarrow \mathrm{argmax}_{\mathbf{v} \in \mathcal{S}_t} \langle \nabla f(\mathbf{x}_t), \mathbf{v} \rangle$
5:      **if** $\langle \nabla f(\mathbf{x}_t), \mathbf{x}_t - \mathbf{v}_t \rangle \geq \langle \nabla f(\mathbf{x}_t), \mathbf{a}_t - \mathbf{x}_t \rangle$ **then**
6:          $\mathbf{d}_t \leftarrow \mathbf{v}_t - \mathbf{x}_t$, $\gamma_{\max} \leftarrow 1$
7:      **else**
8:          $\mathbf{d}_t \leftarrow \mathbf{x}_t - \mathbf{a}_t$, $\gamma_{\max} \leftarrow \boldsymbol{\lambda}_t(\mathbf{a}_t)/(1 - \boldsymbol{\lambda}_t(\mathbf{a}_t))$
9:      $\gamma_t, L_t \leftarrow \mathtt{Backtrack}(f, \mathbf{x}_t, \mathbf{d}_t, \nabla f(\mathbf{x}_t), L_{t-1}, \gamma_{\max})$
10:     $\mathbf{x}_{t+1} \leftarrow \mathbf{x}_t + \gamma_t \mathbf{d}_t$
11:     Update $\mathcal{S}_t$ and $\boldsymbol{\lambda}_t$ to $\mathcal{S}_{t+1}$ and $\boldsymbol{\lambda}_{t+1}$

---

The proof of linear convergence follows closely from Pedregosa et al. (2020) and Lacoste-Julien & Jaggi (2015), with the only difference that we need to take into consideration that the function is generalized self-concordant as opposed to smooth and strongly convex. One of the key inequalities used in the proof is a scaling inequality from Lacoste-Julien & Jaggi (2015) very similar to the one shown in Proposition A.4 and Proposition A.7, which we state next:

**Proposition B.1.** *Let $\mathcal{X} \subseteq \mathbb{R}^n$ be a polytope, and denote by $\mathcal{S}$ the set of vertices of the polytope $\mathcal{X}$ that gives rise to $\mathbf{x} \in \mathcal{X}$ as a convex combination with positive weights, then for all $\mathbf{y} \in \mathcal{X}$:*

$$\langle \nabla f(\mathbf{x}), \mathbf{a} - \mathbf{v} \rangle \geq \delta \frac{\langle \nabla f(\mathbf{x}), \mathbf{x} - \mathbf{y} \rangle}{\|\mathbf{x} - \mathbf{y}\|},$$

*where $\mathbf{v} = \mathrm{argmin}_{\mathbf{u} \in \mathcal{X}} \langle \nabla f(\mathbf{x}), \mathbf{u} \rangle$, $\mathbf{a} = \mathrm{argmax}_{\mathbf{u} \in \mathcal{S}} \langle \nabla f(\mathbf{x}), \mathbf{u} \rangle$, and $\delta > 0$ is the* pyramidal width *of $\mathcal{X}$.*

**Theorem B.2.** *Suppose $\mathcal{X}$ is a polytope and $f$ is a $(M, \nu)$ generalized self-concordant function with $\nu \geq 2$ for which the domain does not contain straight lines. Then the Away-step Frank-Wolfe*

*(AFW) algorithm with Backtrack (Algorithm 7) results in a convergence:*

$$h(\mathbf{x}_t) \leq h(\mathbf{x}_0) \left( 1 - \frac{\mu_f^{\mathcal{L}_0}}{4\tilde{L}} \left( \frac{\delta}{D} \right)^2 \right)^{\lceil (t-1)/2 \rceil},$$

*where $\delta$ is the pyramidal width of the polytope $\mathcal{X}$, $\tilde{L} \overset{\text{def}}{=} \max\{\tau L_f^{\mathcal{L}_0}, L_{-1}\}$, $\tau > 1$ is the backtracking parameter, $L_{-1}$ is the initial smoothness estimate in Algorithm 6, $\mu_f^{\mathcal{L}_0} = \min_{\mathbf{u} \in \mathcal{L}_0, \mathbf{d} \in \mathbb{R}^n} \|\mathbf{d}\|_{\nabla^2 f(\mathbf{u})}^2 / \|\mathbf{d}\|_2^2$ and $L_f^{\mathcal{L}_0} = \max_{\mathbf{u} \in \mathcal{L}_0, \mathbf{d} \in \mathbb{R}^n} \|\mathbf{d}\|_{\nabla^2 f(\mathbf{u})}^2 / \|\mathbf{d}\|_2^2$.*

*Proof.* Proceeding very similarly as in the proof of Theorem A.5, we have that as the backtracking line search makes monotonous primal progress, we know that for $t \geq 0$ we will have that $\mathbf{x}_t \in \mathcal{L}_0$. As the function is $\mu_f^{\mathcal{L}_0}$-strongly convex over $\mathcal{L}_0$, we can use the appropriate inequalities from strong convexity in the progress bounds. Using this aforementioned property, together with the scaling inequality of Proposition B.1 results in:

$$f(\mathbf{x}_t) - f(\mathbf{x}^*) \leq \frac{\langle \nabla f(\mathbf{x}_t), \mathbf{x}_t - \mathbf{x}^* \rangle}{2\mu_f^{\mathcal{L}_0} \|\mathbf{x}_t - \mathbf{x}^*\|^2} \tag{B.1}$$

$$\leq \frac{\langle \nabla f(\mathbf{x}_t), \mathbf{a}_t - \mathbf{v}_t \rangle^2}{2\mu_f^{\mathcal{L}_0} \delta^2} \tag{B.2}$$

$$= \frac{(\langle \nabla f(\mathbf{x}_t), \mathbf{a}_t - \mathbf{x}_t \rangle + \langle \nabla f(\mathbf{x}_t), \mathbf{x}_t - \mathbf{v}_t \rangle)^2}{2\mu_f^{\mathcal{L}_0} \delta^2}, \tag{B.3}$$

where the first inequality comes from the $\mu_f^{\mathcal{L}_0}$-strong convexity over $\mathcal{L}_0$, and the second inequality comes from applying Proposition B.1 with $\mathbf{y} = \mathbf{x}^*$. Note that if the Frank-Wolfe step is chosen in Line 6, then $-\langle \nabla f(\mathbf{x}_t), \mathbf{d}_t \rangle = \langle \nabla f(\mathbf{x}_t), \mathbf{x}_t - \mathbf{v}_t \rangle \geq \langle \nabla f(\mathbf{x}_t), \mathbf{a}_t - \mathbf{x}_t \rangle$, otherwise, if an away step is chosen in Line 8, then $\langle \nabla f(\mathbf{x}_t), \mathbf{x}_t - \mathbf{v}_t \rangle < \langle \nabla f(\mathbf{x}_t), \mathbf{a}_t - \mathbf{x}_t \rangle = -\langle \nabla f(\mathbf{x}_t), \mathbf{d}_t \rangle$, in any case, plugging into Equation (B.3) we have that:

$$h(\mathbf{x}_t) = f(\mathbf{x}_t) - f(\mathbf{x}^*) \leq \frac{2 \langle \nabla f(\mathbf{x}_t), \mathbf{d}_t \rangle^2}{\mu_f^{\mathcal{L}_0} \delta^2}. \tag{B.4}$$

Note that using a similar reasoning, as $\langle \nabla f(\mathbf{x}_t), \mathbf{x}_t - \mathbf{v}_t \rangle = g(\mathbf{x}_t)$, in both cases it holds that:

$$h(\mathbf{x}_t) \leq g(\mathbf{x}_t) \leq -\langle \nabla f(\mathbf{x}_t), \mathbf{d}_t \rangle. \tag{B.5}$$

As in the preceding proofs, the backtracking line search in Algorithm 6 will either output a point $\gamma_t = \gamma_{\max}$ or $\gamma_t < \gamma_{\max}$. In any case, and regardless of if a Frank-Wolfe step (Line 6) or an away step (Line 8) is chosen, Algorithm 6 will find and output a smoothness estimate $L_t$ and a step size $\gamma_t$ such that:

$$h(\mathbf{x}_{t+1}) - h(\mathbf{x}_t) \leq \frac{L_t \gamma_t^2}{2} \|\mathbf{d}_t\|^2 + \gamma_t \langle \nabla f(\mathbf{x}_t), \mathbf{d}_t \rangle. \tag{B.6}$$

As before, we will have two different cases. If $\gamma_t = \gamma_{\max}$ we know by observing Line 5 of Algorithm 6 that $-\langle \nabla f(\mathbf{x}_t), \mathbf{d}_t \rangle \geq \gamma_{\max} L_t \|\mathbf{d}_t\|^2$, and so plugging into Equation (B.6) we arrive at $h(\mathbf{x}_{t+1}) - h(\mathbf{x}_t) \leq \langle \nabla f(\mathbf{x}_t), \mathbf{d}_t \rangle \gamma_{\max}/2$. In the case where $\gamma_t < \gamma_{\max}$, we have that $-\langle \nabla f(\mathbf{x}_t), \mathbf{d}_t \rangle < \gamma_{\max} L_t \|\mathbf{d}_t\|^2$ and $\gamma_t = -\langle \nabla f(\mathbf{x}_t), \mathbf{d}_t \rangle / (L_t \|\mathbf{d}_t\|^2)$, and so plugging into Equation (B.6) we arrive at $h(\mathbf{x}_{t+1}) - h(\mathbf{x}_t) \leq -\langle \nabla f(\mathbf{x}_t), \mathbf{d}_t \rangle^2 / (2L_t \|\mathbf{d}_t\|^2)$. In any case, we can rewrite Equation (B.6) as:

$$h(\mathbf{x}_t) - h(\mathbf{x}_{t+1}) \geq \min \left\{ \frac{-\langle \nabla f(\mathbf{x}_t), \mathbf{d}_t \rangle \gamma_{\max}}{2}, \frac{\langle \nabla f(\mathbf{x}_t), \mathbf{d}_t \rangle^2}{2L_t \|\mathbf{d}_t\|^2} \right\}. \tag{B.7}$$

We can now use the inequality in Equation (B.4) to bound the second term in the minimization component of Equation (B.7), and Equation (B.5) to bound the first term. This leads to:

$$h(\mathbf{x}_t) - h(\mathbf{x}_{t+1}) \geq h(\mathbf{x}_t) \min\left\{\frac{\gamma_{\max}}{2}, \frac{\mu_f^{\mathcal{L}_0}\delta^2}{4L_t\|\mathbf{d}_t\|^2}\right\} \tag{B.8}$$

$$\geq h(\mathbf{x}_t) \min\left\{\frac{\gamma_{\max}}{2}, \frac{\mu_f^{\mathcal{L}_0}\delta^2}{4\tilde{L}D^2}\right\}. \tag{B.9}$$

where in the last inequality we have used $\|d_t\| \leq D$ and $L_t \leq \tilde{L}$ for all $t$. It remains to bound $\gamma_{\max}$ away from zero to obtain the linear convergence bound. For Frank-Wolfe steps we immediately have $\gamma_{\max} = 1$, but for away steps there is no straightforward way of bounding $\gamma_{\max}$ away from zero. One of the key insights from Lacoste-Julien & Jaggi (2015) is that instead of bounding $\gamma_{\max}$ away from zero for all steps up to iteration $t$, we can instead bound the number of away steps with a step size $\gamma_t = \gamma_{\max}$ up to iteration $t$, which are steps that reduce the cardinality of the active set $\mathcal{S}_t$ and satisfy $h(\mathbf{x}_t) \leq h(\mathbf{x}_{t+1})$. This leads us to consider only the progress provided by the remaining steps, which are away steps with $\gamma_t < \gamma_{\max}$, and Frank-Wolfe steps. For a number of steps $t$, only at most half of these steps could have been away steps with $\gamma_t = \gamma_{\max}$, as we cannot drop more vertices from the active set than the number of vertices we could have potentially picked up with Frank-Wolfe steps. For the remaining $\lceil(t-1)/2\rceil$ steps we know that $h(\mathbf{x}_t) - h(\mathbf{x}_{t+1}) \geq h(\mathbf{x}_t)\mu_f^{\mathcal{L}_0}\delta^2/(4\tilde{L}D^2)$. Therefore we have that the primal gap satisfies:

$$h(\mathbf{x}_t) \leq h(\mathbf{x}_0)\left(1 - \frac{\mu_f^{\mathcal{L}_0}\delta^2}{4\tilde{L}D^2}\right)^{\lceil(t-1)/2\rceil}.$$

This completes the proof. □

We can make use of the proof of convergence in primal gap to prove linear convergence in Frank-Wolfe gap. In order to do so, we recall a quantity formally defined in Kerdreux et al. (2019) but already implicitly used earlier in Lacoste-Julien & Jaggi (2015) as:

$$w(\mathbf{x}_t, \mathcal{S}_t) \stackrel{\text{def}}{=} \max_{\mathbf{u}\in\mathcal{S}_t, \mathbf{v}\in\mathcal{X}} \langle\nabla f(\mathbf{x}_t), \mathbf{u} - \mathbf{v}\rangle$$

$$= \max_{\mathbf{u}\in\mathcal{S}_t} \langle\nabla f(\mathbf{x}_t), \mathbf{u} - \mathbf{x}_t\rangle + \max_{\mathbf{v}\in\mathcal{X}} \langle\nabla f(\mathbf{x}_t), \mathbf{x}_t - \mathbf{v}\rangle$$

$$= \max_{\mathbf{u}\in\mathcal{S}_t} \langle\nabla f(\mathbf{x}_t), \mathbf{u} - \mathbf{x}_t\rangle + g(\mathbf{x}_t).$$

Note that as the first term, the so-called *away gap* in the previous equation is positive and hence $w(\mathbf{x}_t, \mathcal{S}_t)$ provides an upper bound on the Frank-Wolfe gap.

**Theorem B.3.** *Suppose $\mathcal{X}$ is a polytope and $f$ is a $(M, \nu)$ generalized self-concordant function with $\nu \geq 2$ for which the domain does not contain straight lines. Then the Away-step Frank-Wolfe (AFW) algorithm with Backtrack (Algorithm 7) contracts the Frank-Wolfe gap linearly, i.e., $\min_{1\leq t\leq T} g(\mathbf{x}_t) \leq \varepsilon$ after $T = O(\log 1/\varepsilon)$ iterations.*

*Proof.* Note that the condition in Line 5 of Algorithm 7 means that regardless of if we chose to perform an away step of a Frank-Wolfe step, we have that $-2\langle\nabla f(\mathbf{x}_t), \mathbf{d}_t\rangle \geq \langle\nabla f(\mathbf{x}_t), \mathbf{x}_t - \mathbf{v}_t\rangle + \langle\nabla f(\mathbf{x}_t), \mathbf{a}_t - \mathbf{x}_t\rangle = w(\mathbf{x}_t, \mathcal{S}_t)$. On the other hand, we also have that $h(\mathbf{x}_t) - h(\mathbf{x}_{t+1}) \leq h(\mathbf{x}_t)$. Plugging these bounds into the right-hand side and the left hand side of Equation B.7 in Theorem B.2, and using the fact that $\|\mathbf{d}_t\| \leq D$ we have that:

$$\min\left\{\frac{w(\mathbf{x}_t, \mathcal{S}_t)\gamma_{\max}}{4}, \frac{w(\mathbf{x}_t, \mathcal{S}_t)^2}{8L_tD^2}\right\} \leq h(\mathbf{x}_t) \leq h(\mathbf{x}_0)\left(1 - \frac{\mu_f^{\mathcal{L}_0}}{4\tilde{L}}\left(\frac{\delta}{D}\right)^2\right)^{\lceil(t-1)/2\rceil},$$

where the second inequality follows from the convergence bound on the primal gap from Theorem B.2. Considering the steps that are not away steps with $\gamma_t = \gamma_{\max}$ as in the proof of Theorem B.2, leads us to:

$$g(\mathbf{x}_t) \leq w(\mathbf{x}_t, \mathcal{S}_t) \leq 4h(\mathbf{x}_0) \max\left\{1, \sqrt{\frac{\tilde{L}D^2}{2h(\mathbf{x}_0)}}\right\} \left(1 - \frac{\mu_f^{\mathcal{L}_0}}{4\tilde{L}}\left(\frac{\delta}{D}\right)^2\right)^{\lfloor(t-1)/4\rfloor}.$$

$\square$

In Table 4 we provide a detailed complexity comparison between the Backtracking AFW algorithm (`B-AFW`) algorithm, which can be found in Algorithm 7 in the appendix, and other comparable algorithms in the literature. Note that these complexities assume that the domain under consideration is polyhedral.

| Algorithm | SOO calls | FOO calls | ZOO calls | LMO calls | DO calls |
|---|---|---|---|---|---|
| `FW-LLOO` [1, Alg.7] | $O(\log 1/\varepsilon)$ | $O(\log 1/\varepsilon)$ | | $O(\log 1/\varepsilon)$* | |
| `ASFW-GSC` [1, Alg.8] | $O(\log 1/\varepsilon)$ | $O(\log 1/\varepsilon)$ | | $O(\log 1/\varepsilon)$ | |
| `B-AFW`[†‡] [**This work**] | | $O(\log 1/\varepsilon)$ | $O(\log 1/\varepsilon)$ | $O(\log 1/\varepsilon)$ | $O(\log 1/\varepsilon)$ |

Table 4: **Complexity comparison**: Number of iterations needed to reach an $\varepsilon$-optimal solution in $h(\mathbf{x})$ for Problem 1.1. We denote Dvurechensky et al. (2020b) by [1]. The asterisk on `FW-LLOO` highlights the fact that the oracle is different from the standard `LMO`. Lastly, the complexity shown for the `FW-LLOO`, `ASFW-GSC`, and `B-AFW` algorithms only apply to polyhedral domains, with the additional requirement that for the former two we need an explicit polyhedral representation of the domain. We use the superscript † to indicate that the same complexities hold when reaching an $\varepsilon$-optimal solution in $g(\mathbf{x})$, and the superscript ‡ to indicate that constants in the convergence bounds depend on user-defined inputs.

## C  Remarks and supplementary information on the experimental section

We ran all experiments on a server with 8 Intel Xeon 3.50GHz CPUs and 32GB RAM. All computations are run in single-threaded mode using Julia 1.6.0 with the `FrankWolfe.jl` package. The data sets used in the problem instances can be found in Carderera et al. (2021), the code used for the experiments can be found on https://github.com/ZIB-IOL/fw-generalized-selfconcordant. When running the adaptive step size from Pedregosa et al. (2020), the only parameter that we need to set is the initial smoothness estimate $L_{-1}$. We use the initialization proposed in the Pedregosa et al. (2020) paper, namely:

$$L_{-1} = \|\nabla f(\mathbf{x}_0) - \nabla f(\mathbf{x}_0 + \varepsilon(\mathbf{v}_0 - \mathbf{x}_0))\| / (\varepsilon \|\mathbf{v}_0 - \mathbf{x}_0\|)$$

with $\varepsilon$ set to $10^{-3}$. The scaling parameters $\tau = 2, \eta = 0.9$ are left at their default values as proposed in Pedregosa et al. (2020) and also used in Dvurechensky et al. (2020b).

We provide the full details of the experiments carried out in the paper:

**Portfolio optimization.**  We consider $f(\mathbf{x}) = -\sum_{t=1}^{p} \log(\langle \mathbf{r}_t, \mathbf{x}\rangle)$, where $p$ denotes the number of periods and $\mathcal{X} = \Delta_n$. The results are shown in Figure 2. We use the revenue data $\mathbf{r}_t$ from Dvurechensky et al. (2020b) and add instances generated in a similar fashion from independent Normal random entries with 1000, 2000, and 5000 dimensions, and from a Log-normal distribution with ($\mu = 0.0, \sigma = 0.5$).

**Signal recovery with KL divergence.**  We apply the aforementioned algorithms to the recovery of a sparse signal from a noisy linear image using the Kullback-Leibler divergence. Given a linear map $W$, we assume a signal $\mathbf{y}$ is generated by $\mathbf{y} = W\mathbf{x}_0 + \varepsilon$, where $\mathbf{x}_0$ is assumed to be a sparse unknown input signal and $\varepsilon$ is a random error. Assuming $W$ and $\mathbf{y}$ are entrywise positive, and that the signal to recover should also be entrywise positive, the

minimizer of the KL divergence (or Kullback's I-divergence (Csiszar, 1991)) can be used as an estimator for $\mathbf{x}_0$. The KL divergence between the resulting output signals is expressed as $f(\mathbf{x}) = D(W\mathbf{x}, \mathbf{y}) = \sum_{i=1}^{N} \left\{ \langle \mathbf{w}_i, \mathbf{x} \rangle \log \left( \frac{\langle \mathbf{w}_i, \mathbf{x} \rangle}{y_i} \right) - \langle \mathbf{w}_i, \mathbf{x} \rangle + y_i \right\}$, where $\mathbf{w}_i$ is the $i^{\text{th}}$ row of $W$. In order to promote sparsity and enforce nonnegativity of the solution, we use the unit simplex of radius $R$ as the feasible set $\mathcal{X} = \{\mathbf{x} \in \mathbb{R}_+^d, \|\mathbf{x}\|_1 \leq R\}$. The results are shown in Figure 3. We used the same $M = 1$ choice for the second-order method as in Dvurechensky et al. (2020b) for comparison; whether this choice is admissible is unknown (see Remark C.1). We generate input signals $\mathbf{x}_0$ with 30% non-zeros elements following an exponential distribution of mean $\lambda = 1$. The entries of $W$ are generated from a folded Normal distribution built from absolute values of Gaussian random numbers with standard deviation 5 and mean 0. The additive noise is generated from a Gaussian centered distribution with a standard deviation equal to a fraction of the standard deviation of $W\mathbf{x}_0$.

**Logistic regression.** One of the motivating examples for the development of a theory of generalized self-concordant function is the logistic regression problem, as it does not match the definition of a standard self-concordant function but shares many of its characteristics. We consider a design matrix with rows $\mathbf{a}_i \in \mathbb{R}^n$ with $1 \leq i \leq N$ and a vector $\mathbf{y} \in \{-1, 1\}^N$ and formulate a logistic regression problem with elastic net regularization, in a similar fashion as is done in Liu et al. (2020), with $f(\mathbf{x}) = 1/N \sum_{i=1}^{N} \log(1 + \exp(-y_i \langle \mathbf{x}, \mathbf{a}_i \rangle)) + \mu/2 \|\mathbf{x}\|^2$, and $\mathcal{X}$ is the $\ell_1$ ball of radius $\rho$, where $\mu$ and $\rho$ are two regularization parameters. The logistic regression loss is generalized self-concordant with $\nu = 2$. The results can be seen in Figure 4 and expanded in Section C. We use the a1a-a9a datasets from the LIBSVM classification data.

**Birkhoff polytope.** All applications previously considered all have in common a constraint set possessing computationally inexpensive LMOs (probability or unit simplex and $\ell_1$ norm ball). Additionally, each vertex returned from the LMO is highly sparse with at most one non-zero element. To complement the results we consider a problem over the Birkhoff polytope, the polytope of doubly stochastic matrices, where the LMO is implemented through the Hungarian algorithm, and is not as inexpensive as in the other examples considered. We use a quadratic regularization parameter $\mu = 100/\sqrt{n}$ where $n$ is the number of samples.

**Remark C.1.** Note that Proposition 2 in Sun & Tran-Dinh (2019), which deals with the composition of generalized self-concordant functions with affine maps, does not apply to the KL divergence objective function, reproduced here for reference:

$$f(\mathbf{x}) = D(W\mathbf{x}, \mathbf{y}) = \sum_{i=1}^{N} \left\{ \langle \mathbf{w}_i, \mathbf{x} \rangle \log \left( \frac{\langle \mathbf{w}_i, \mathbf{x} \rangle}{y_i} \right) - \langle \mathbf{w}_i, \mathbf{x} \rangle + y_i \right\}.$$

Furthermore, the objective function is strongly convex if and only if $\text{rank}(W) \geq n$, where $n$ is the dimension of the problem.

*Proof.* (Sun & Tran-Dinh, 2019, Proposition 2) establishes certain conditions under which the composition of a generalized self-concordant function with an affine map results in a generalized self-concordant function. The objective is of the form:

$$\sum_{i=1}^{N} \phi_i(\langle \mathbf{w}_i, \mathbf{x} \rangle) \quad \text{with} \quad \phi_i(t) = t \log \left( \frac{t}{y_i} \right) - t + y_i = t \log t - t \log y_i - t + y_i.$$

Note that generalized self-concordant functions are closed under addition, and so we only focus on the individual terms in the sum. As first-order terms are $(0, \nu)$-generalized self-concordant for any $\nu > 0$, then we know that the composition of these first-order terms with an affine map results in a generalized self-concordant function Sun & Tran-Dinh (2019, Proposition 2). We therefore focus on the entropy function $t \log t$ which is $(1, 4)$ generalized self-concordant. The conditions which ensure that the composition of a $(M, \nu)$-generalized self-concordant function with an affine map $x \mapsto Ax$ results in a generalized self-concordant function requires in the case $\nu > 3$ that $\lambda_{min}(A^T A) > 0$ (Sun & Tran-Dinh, 2019, Proposition 2). In the case of the KL divergence objective, $A = \mathbf{w}_i^T$ and $A^T A = \mathbf{w}_i \mathbf{w}_i^T$ is an outer product with only one positive eigenvalue, and 0 of multiplicity $n - 1$. Therefore we cannot guarantee

that the function $\phi_i(\langle \mathbf{w}_i, \mathbf{x} \rangle)$ is generalized self-concordant by application of Proposition 2 in Sun & Tran-Dinh (2019).

Alternatively, in order to try to show that the function is generalized self-concordant we could consider $f(\mathbf{x}) := g(W\mathbf{x})$. Assuming rank$(W) \geq n$, then $W^T W$ is positive definite, and only the generalized self-concordance of $g$ is left to prove.

$$g(\mathbf{z}) = \sum_{i=1}^{N} \phi_i(\mathbf{z}_i).$$

Each term $\mathbf{z} \mapsto \phi_i(\mathbf{z}_i) = \phi_i(\mathbf{e}_i^T \mathbf{z})$ with $\mathbf{e}_i$ the $i^{\text{th}}$ standard basis vector is the composition of a generalized self-concordant function composed with a rank-one affine transformation, this raises the same issues encountered in the paragraph above.

Regarding the strong-convexity of the objective function, we can express the gradient and the Hessian of the function as:

$$\nabla f(\mathbf{x}) = \sum_{i=1}^{N} \mathbf{w}_i (\log \langle \mathbf{w}_i, \mathbf{x} \rangle - \log y_i)$$

$$\nabla^2 f(\mathbf{x}) = \sum_{i=1}^{N} \frac{\mathbf{w}_i \mathbf{w}_i^T}{\langle \mathbf{w}_i, \mathbf{x} \rangle},$$

which is the sum of $N$ outer products, each corresponding to a single eigenvector $\mathbf{w}_i$. If rank$(W) \geq n$, the Hessian is definite positive and the objective is strongly convex. Otherwise, it possesses zero as an eigenvalue regardless of $\mathbf{x}$, and the function Hessian is positive semi-definite. □

**Strong convexity parameter for the LLOO.** The LLOO procedure explicitly requires a strong convexity parameter $\sigma_f$ of the objective function, an underestimator of $\lambda_{min}(\nabla^2 f(\mathbf{x}))$ over $\mathcal{X}$. For the portfolio optimization problem, the Hessian is a sum of rank-one terms:

$$\nabla^2 f(\mathbf{x}) = \sum_t \frac{\mathbf{r}_t \mathbf{r}_t^T}{\langle \mathbf{r}_t, \mathbf{x} \rangle}.$$

The only non-zero eigenvalue associated with each $t$ term is bounded below over $\mathcal{X}$ by:

$$\frac{\|\mathbf{r}_t\|^2}{\max_{\mathbf{x} \in \mathcal{X}} \langle \mathbf{r}_t, \mathbf{x} \rangle^2} = \frac{\|\mathbf{r}_t\|^2}{\max\{\max_{\mathbf{x} \in \mathcal{X}} \langle \mathbf{r}_t, \mathbf{x} \rangle, -\min_{\mathbf{x} \in \mathcal{X}} \langle \mathbf{r}_t, \mathbf{x} \rangle\}^2}.$$

The denominator can be solved by two calls to the LMO, and we will denote it by $\beta_t$ for the $t^{\text{th}}$ term. Each summation term contributes positively to one of the eigenvalues of the Hessian matrix, an underestimator of the the strong convexity parameter is then given by:

$$\lambda_{min} \left( \sum_t \frac{\mathbf{r}_t \mathbf{r}_t^T}{\beta_t} \right).$$

The second-order method `GSC-FW` has been implemented with an in-place Hessian matrix updated at each iteration, following the implementation of Dvurechensky et al. (2020b). The Hessian computation nonetheless adds significant cost in the runtime of each iteration, even if the local norm and other quadratic expressions $\langle \nabla^2 f(\mathbf{x})\mathbf{y}, \mathbf{z} \rangle$ can be computed allocation-free. A potential improvement for future work would be to represent Hessian matrices as functional linear operators mapping any $\mathbf{y}$ to $\nabla^2 f(\mathbf{x})\mathbf{y}$.

**Monotonous step size: the numerical case.** The computational experiments highlighted that the Monotonous Frank-Wolfe performs well in terms of iteration count and time against other Frank-Wolfe and Away-step Frank-Wolfe variants. Another advantage of a simple step size computation procedure is its numerical stability. On some instances, an ill-conditioned gradient can lead to a plateau of the primal and/or dual progress. Even worse, some step-size

strategies do not guarantee monotonicity and can result in the primal value increasing over some iterations. The numerical issue that causes this phenomenon is illustrated by running the methods of the `FrankWolfe.jl` package over the same instance using 64-bits floating-point numbers and Julia `BigFloat` types (which support arithmetic in arbitrary precision to remove numerical issues).

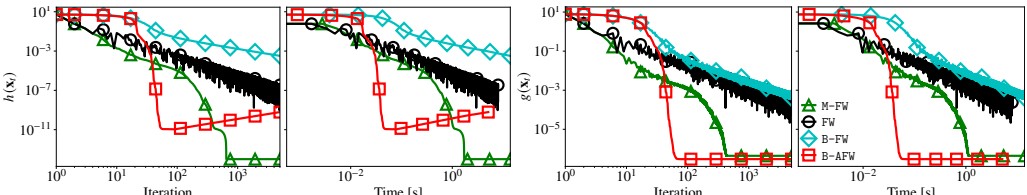

Figure 6: Ill-conditioned portfolio optimization problem solved using floating-point arithmetic.

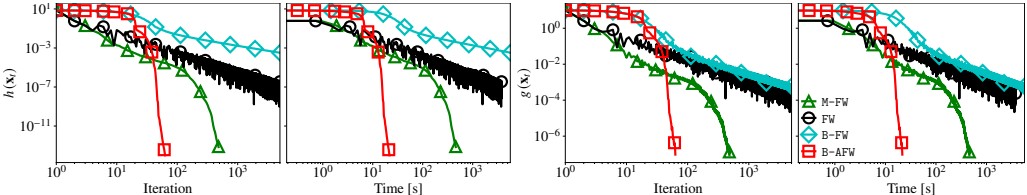

Figure 7: Ill-conditioned portfolio optimization problem solved using arbitrary precision.

In Figure 6, we observe a plateau of the dual gap for both `M-FW` and `B-AFW`. The primal value however worsens after the iteration where `B-AFW` reaches its dual gap plateau. In contrast, `M-FW` reaches a plateau in both primal and dual gap at a certain iteration. Note that the primal value at the point where the plateau is hit is already below $\sqrt{\varepsilon_{\text{float64}}}$, the square root of the machine precision. The same instance and methods operating in arbitrary precision arithmetic are presented Figure 7. Instead of reaching a plateau or deteriorating, `B-AFW` closes the dual gap tolerance and terminates before other methods. Although this observation (made on several instances of the portfolio optimization problem) only impacts ill-conditioned problems, it suggests `M-FW` may be a good candidate for a numerically robust default implementation of Frank-Wolfe algorithms.

**Function domain in the constraints**   One of the arguments used to motivate the construction of FW algorithms for standard and generalized self-concordant minimization in prior work is the difficulty of handling objective functions with implicitly defined domains. We make several observations that highlight the relevance of this issue and justify the assumption of the availability of a Domain Oracle for $\text{dom}(f)$. In the portfolio optimization example, all revenue vectors are assumed positive, $\mathbf{r}_t \in \mathbb{R}^n_{++}$ and $\mathbf{x} \in \Delta_n$, it follows that all feasible points lie in $\text{dom}(f)$. More generally ,for the logarithm of an affine function, verifying that a candidate $\mathbf{x}$ lies in $\text{dom}(f)$ consists of a single affine transformation and element-wise comparison $A\mathbf{x} + \mathbf{b} > \mathbb{R}^m_{++}$.

In the inverse covariance estimation problem, the information on $\text{dom}(f)$ can be added to the constraints by imposing $\text{mat}(\mathbf{x}) \in \mathbb{S}^n_+$, yielding a semi-definite optimization problem. The domain oracle consists of the computation of the smallest eigenvalue, which needs to be positive.

We can also modify the feasible region of the signal retrieval application using the KL divergence, resulting in a new feasible region $\mathcal{X}'$ so that $\text{Int}(\mathcal{X}') \subset \text{dom}(f)$. The objective is of the form:

$$f(\theta) = \sum_i \langle \mathbf{w}_i, \theta \rangle \log \left( \frac{\langle \mathbf{w}_i, \theta \rangle}{y_i} \right) - \langle \mathbf{w}_i, \theta \rangle$$

where the data $W$ and $y$ are assumed entrywise positive, thus $\text{dom}(f) = \{x \in \mathbb{R}^n_+, x \neq 0\}$. Therefore we can define the set $\mathcal{X}'$ as the unit simplex. The domain of each function involved in the sum in $f$ has an open domain $(0, +\infty)$.

However, the positivity assumption on all these components could be relaxed. Without the positivity assumption on $W$, the Domain Oracle would consist of verifying:

$$\langle \mathbf{w}_i, \theta \rangle > 0 \; \forall i. \tag{C.1}$$

This verification can however be simplified by a preprocessing step if the number of data points is large by finding the minimal set of supporting hyperplanes in the polyhedral cone (C.1), which we can find by solving the following linear problem:

$$\max_{\tau, \theta} \tau \tag{C.2a}$$

$$\text{s.t. } \tau \leq \langle \mathbf{w}_i, \theta \rangle \; \forall i \;\; (\lambda_i) \tag{C.2b}$$

$$\|\theta\|_1 \leq R, \tag{C.2c}$$

where $\lambda_i$ is the dual variable associated with the $i^{\text{th}}$ inner product constraint. If the optimal solution of Problem (C.2a) is 0, the original problem is infeasible and the cone defined by (C.1) is empty. Otherwise the optimal $\theta$ will lie in the intersection of the closure of the polyhedral cone and the $\ell_1$ norm ball. Furthermore, the support of $\lambda$ provides us with the non-redundant inequalities of the cone. Let $\hat{W}$ be the matrix formed with the rows $\mathbf{w}_i$ such that $\lambda_i > 0$, then the Domain Oracle can be simplified to the verification that $\hat{W}\theta \in \mathbb{R}^n_{++}$. The Distance Weighted Discrimination (DWT) model also considered in Dvurechensky et al. (2020b) was initially presented in Marron et al. (2007), the denominator of each sum element $\xi_i$ is initially constrained to be nonnegative, which makes $\text{Int}(\mathcal{X}) \subseteq \text{dom}(f)$ hold. Even without this additional constraint, the nonnegativity of all $\xi_i$ can be ensured with a minimum set of linear constraints in a fashion similar to the signal retrieval application, thus simplifying the Domain Oracle.

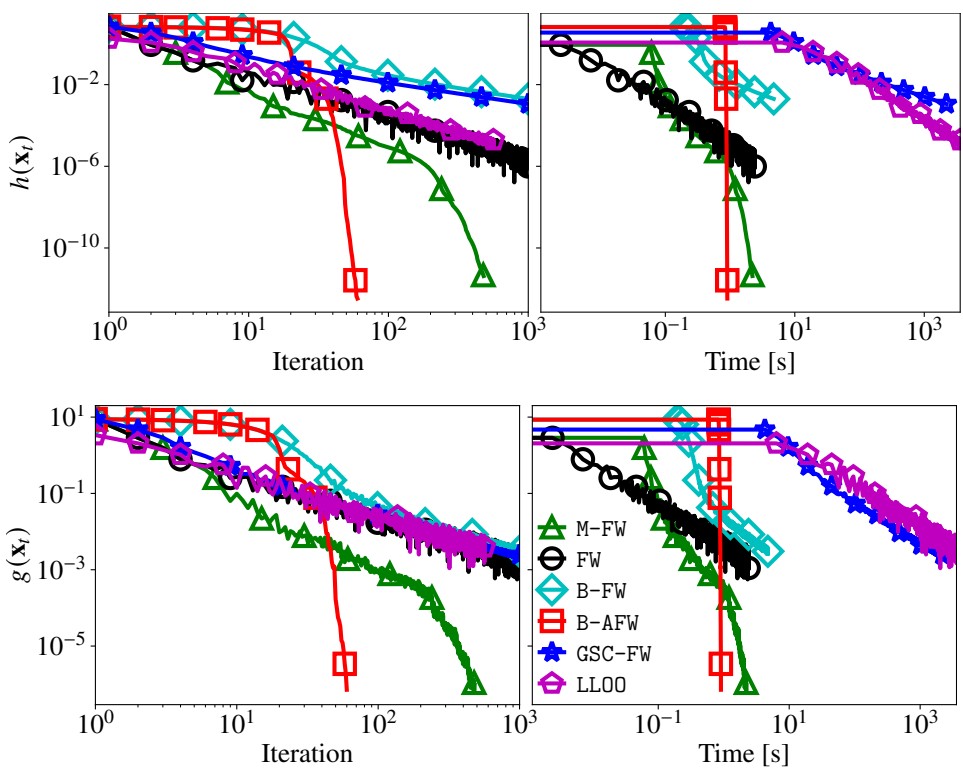

Figure 8: **Portfolio Optimization**: Convergence of $h(\mathbf{x}_t)$ and $g(\mathbf{x}_t)$ vs. $t$ and wall-clock time.

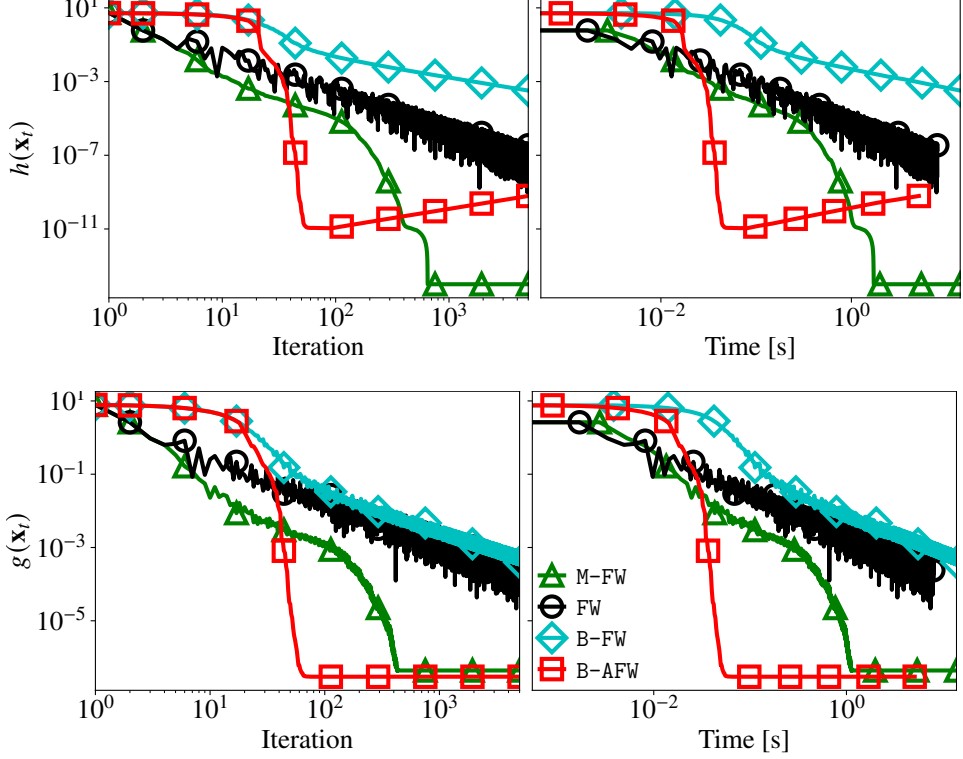

Figure 9: **Portfolio Optimization**: Convergence of $h(\mathbf{x}_t)$ and $g(\mathbf{x}_t)$ vs. $t$ and wall-clock time.

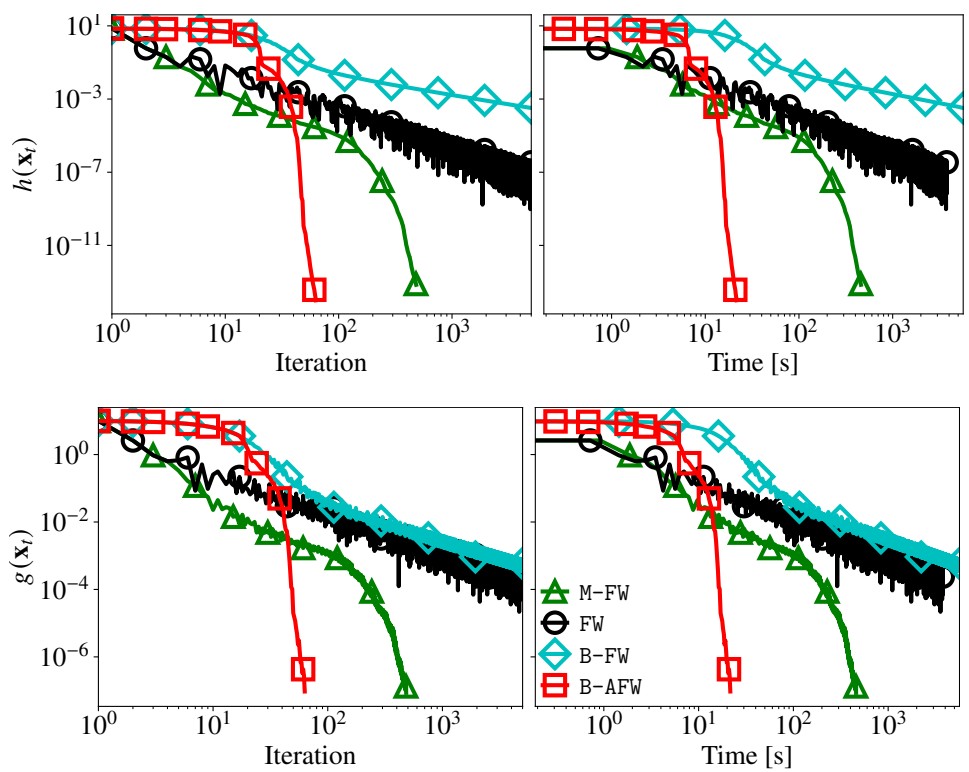

Figure 10: **Portfolio Optimization**: Convergence of $h(\mathbf{x}_t)$ and $g(\mathbf{x}_t)$ vs. $t$ and wall-clock time.

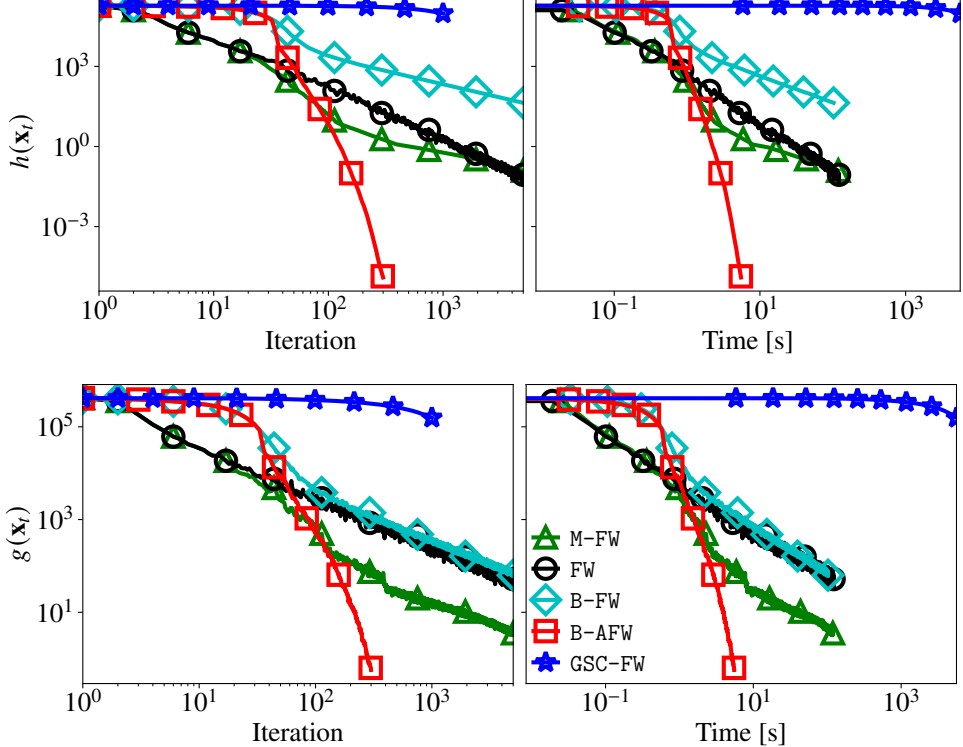

Figure 11: **Signal Recovery**: Convergence of $h(\mathbf{x}_t)$ and $g(\mathbf{x}_t)$ vs. $t$ and wall-clock time.

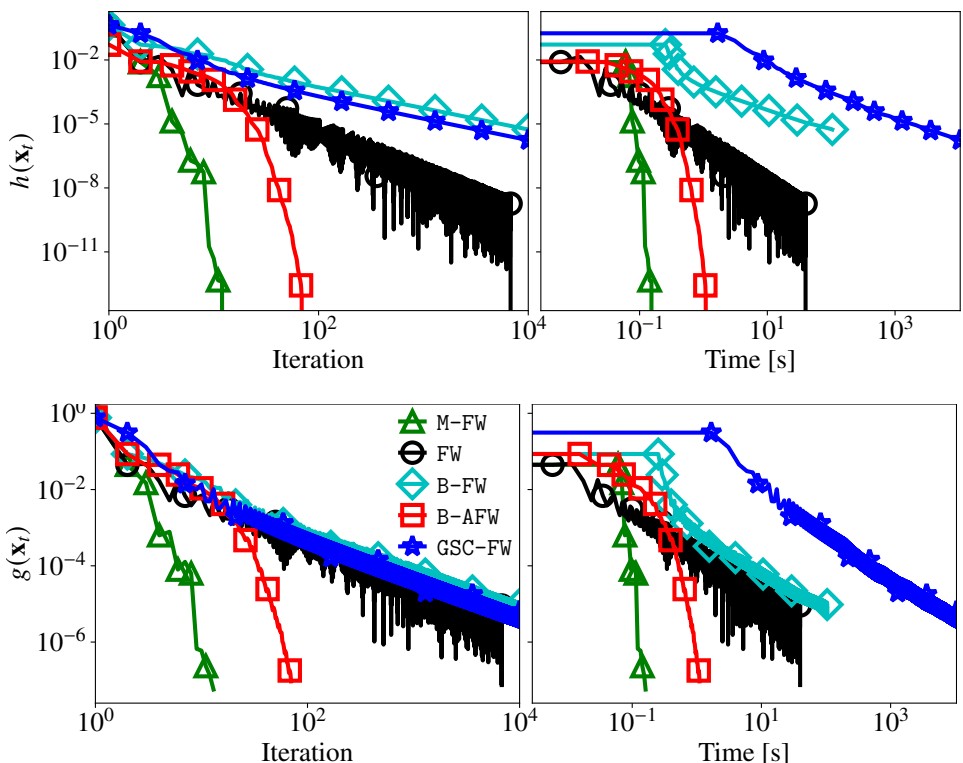

Figure 12: **Logistic Regression**: Convergence of $h(\mathbf{x}_t)$ and $g(\mathbf{x}_t)$ vs. $t$ and wall-clock time for the a4a LIBSVM dataset.

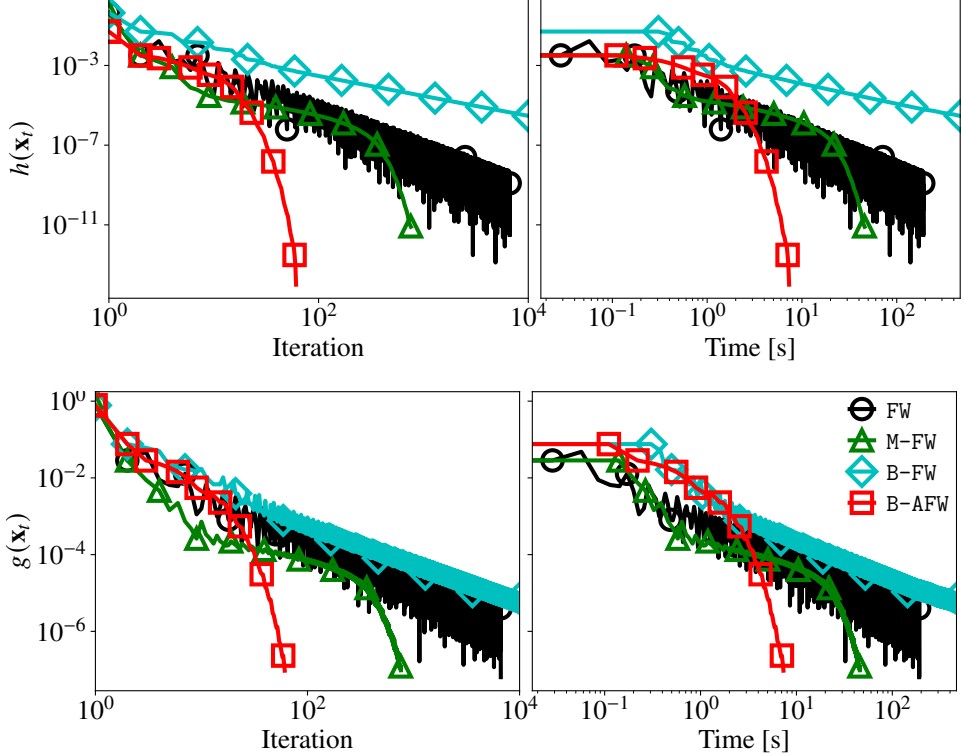

Figure 13: **Logistic Regression**: Convergence of $h(\mathbf{x}_t)$ and $g(\mathbf{x}_t)$ vs. $t$ and wall-clock time for the a8a LIBSVM dataset.

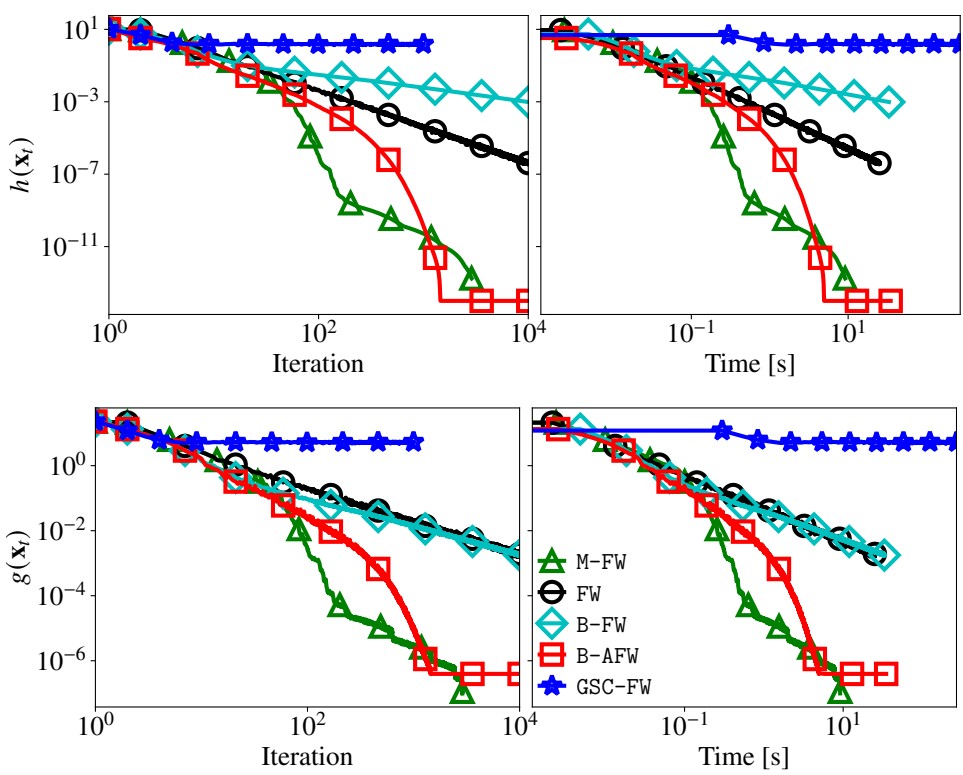

Figure 14: **Birkhoff Polytope**: Convergence of $h(\mathbf{x}_t)$ and $g(\mathbf{x}_t)$ vs. $t$ and wall-clock time.