# OpenReview forum: "Simple steps are all you need: Frank-Wolfe and generalized self-concordant functions"
_NeurIPS.cc/2021/Conference — NeurIPS 2021 Poster_

### Official Review · Reviewer_MQCq · 2021-07-06

**Rating:** 6
**Confidence:** 2

**Summary:**

In this paper, the authors analyse the Conditional Gradient method (Frank-Wolfe algorithm) under the assumption that the objective is a generalized self-concordant function. They propose a simple variant of this method that uses both
1. the universal classical choice of the parameter: $\gamma_t = 2 / (2 + t)$
2. a check for monotonicity
3. a check for domain feasibility

For the method with these settings, the authors establish the global rate of the order O(1/t) in terms of the functional residual and in terms of the duality gap, on the given class of functions.

**Limitations And Societal Impact:**

-

**Main Review:**

The paper is generally well-written. It adresses an interesting and popular research direction related to the analysis of the Conditional Gradient Methods under different assumptions on the problem structure.  In particular, the classes of self-concordant and generalized self-concordant functions are received a substentional attention after the work (Dvurechensky et al. 2020). In the latter paper, there were proposed different step-size strategies for the Frank-Wolfe method that ensure its global convergence. However, these strategies require some additional efforts (such as computation of the Hessian or a line-search).

In the current paper, the author propose a simple and universal stepsize policy ($\gamma_t = 2 / (2 + t)$) which ensures the global rate for the generalized self-concordant functions, when coupled with the monotonicity and feasibility checks. While the combination of these techniques is not novel, the overall setup and new convergence results seem to be interesting for me.

However, I have one major question related to the main result of the paper. For the classical FW algorithm, the known convergence rate is:
$f(x_t) - f^* <= LD^2 / t$
(up to constant factors), where L is the Lipschitz constant for the gradient and D is the diameter of the set.
It looks that the result of Theorem 2.4 provides a bound which is not significantly better than this one.
Instead of $L$, the authors use $L_f$, which is a Lipschitz constant not over the whole set, but only over the initial level set. The payment for such an improvement is an additional factor $T_{\nu}$ and the fact that the new convergence estimate holds for sufficiently big iterations only ($t  \geq T_{\nu}$).
It is not clear to me, how reasonable such result is. What if the value of $T_{\nu}$ is very big (and how big can it be?). Why $L_f$ is much better than $L$, and why it is important for applications.
Maybe, it would be helpful to provide an example when the new estimate is better than the classical one (and what are the values of all the parameters in this case).

Minor remarks:

1. It must be wrong 'et al' in the references: (Bach et al., 2010) and (Nesterov et al., 2018)

2. Line 79: Maybe, it is better not to call the step size $\gamma_t = 2 / (2 + t)$ 'adaptive', since it is predefined in advance and does not depend on the actual objective (does not 'adapt' to its smoothness as for example in the line-search for the gradient methods). For me, it seems to be rather a 'conservative' strategy. I would just propose to pick the most standard terminology.

3. Line 91: 'potentially non-unique': it seems to me that the Hessian is assumed to be strictly positive definite on its domain (to ensure the definition of self-concordance). If so, the function is strictly convex and has a unique minimizer.

4. Line 123: 'Note that if the original objective function is generalized self-concordant, so is the new objective function.' It is not clear to me, why the sum of two generalized self-concordant function (with possibly different $\nu$) is generalized self-concordant. The usual log-barriers are just self-concordant ($\nu$ = 3 is fixed).

5. Line 193: The dot is probably missing after 'the primal progress'.

6. Line 186: 'in a similar fashion as is done in Dvurechensky et al. 2020b ...' -- I would prefer to see the proof of this fact included, or, at least, precise formulation and exact reference (Lemma ??) are given, since it looks like an important part of the proof.

**Time Spent Reviewing:**

4

---

> ### Author Response · Authors · 2021-08-05
> **Response to reviewer MQCq**
>
> We thank the reviewer for the feedback and suggestions, which we believe will help improve the quality of the paper.
>
> We will fix any issues with the bibliography and all small typos in a revised version of the paper. Regarding the specific comments raised by the reviewer:
> 1.	Regarding the comparison to the typical known convergence bound of $f(x_t)−f(x^*)<=LD^2/t$ from the literature, we want to highlight an important aspect. As the function under consideration is generalized self-concordant, the parameter $L$ that defines the smoothness inequality may be infinite/undefined. This is the case for example when the self-concordant function contains a log-barrier used to encode constraints, as in this case the value of $f(x)$ tends to infinity for points $x$ that approach the boundary. This is one of the reasons why the classical $LD^2/t$ is of no use in general for this class of functions, as it does not provide any information, and why the bound in Theorem 2.4 in terms of $L_f$ is useful.
> 2.	Regarding the $T_{\nu}$ parameter in the convergence bounds, it is the cost that one must pay for not using second-order information when computing the step size, and to the best of our knowledge, all existing first-order algorithms for generalized self-concordant functions must pay this price in one form or another (see for example the Dvurechensky et al. paper); in particular line search methods pay this price as well via aggressive backtracking, especially in the beginning. However, compared to (traditional) line search methods, we pay at most one call to zeroth-order, first-order, domain, and LP oracle per iteration rather than e.g., repeated calls to the domain oracle and/or zeroth-order oracle per iteration when backtracking.
> 3.	We thank the reviewer for spotting the typo. In Line 79 we meant to write “agnostic” instead of “adaptive”. Our proposed step size is agnostic in the sense that it is independent of the local smoothness around the current iterate.
> 4.	Regarding Line 91, where we mention that for generalized self-concordant functions the optimum is “potentially non-unique”, note that for example linear functions are a type of generalized self-concordant function (in fact they satisfy the standard definition of self-concordance), and the minimum of a linear function over a polytope can potentially be non-unique. Note also that in general, the definition of generalized self-concordance does not guarantee that the Hessian is positive definite on its domain (to the best of our knowledge “Sun, T. and Tran-Dinh, Q. Generalized self-concordant functions: a recipe for Newton-type methods. Mathematical Programming, 178(1):145–213, 2019” is the only reference we have found that details the additional conditions under which the Hessian of a generalized self-concordant function is positive definite).
> 5.	Regarding Line 123, we thank the reviewer for the question. We will explicitly provide a reference for this fact in the text. See Proposition 1 in “Sun, T. and Tran-Dinh, Q. Generalized self-concordant functions: a recipe for Newton-type methods. Mathematical Programming, 178(1):145–213, 2019”, in which they show that any linear combination with positive weights (in essence, a conical combination) of generalized self-concordant functions is also self-concordant.
> 6.	Regarding the comment about Line 193. We thank the reviewer for carefully reading the manuscript and will reword this sentence to improve the clarity of this part of the text.
> 7.	Regarding the comment in Line 186 – we will add a short proof of the fact we refer to in the paper. In the following, we use $\hat{L_{0}}$ to denote the set defined in Line 184, which is denoted in the text with the mathcal package (we have been unable to get this to render correctly in the response). Basically, as the iterates remain in $\hat{L_{0}}$, and this set is contained inside $\mathrm{dom}(f)$, then the smoothness parameter of the function is bounded over $\hat{L_{0}}$. We denote this smoothness parameter over this domain by $L_{f}^{\hat{L_{0}}}$. This means that as $x_{t} \in \hat{L_{0}}$, which is true by monotonicity, we have that the bound $d^T \nabla^2 f(x_t) d / \lVert d \rVert^2 \leq L_{f}^{\hat{L_{0}}}$ holds for any $d \in \mathbb{R}^n$. This is one of the standard properties of $L_{f}^{\hat{L_{0}}}$-smooth functions. Particularizing for $d = x_t – v_t$ and noting that $\lVert d \rVert \leq D$ we arrive at the desired bound.

---

> > ### Comment · Reviewer_MQCq · 2021-08-22
> > **Response to the answer from authors**
> >
> > Dear Authors,
> >
> > Thank you very much for your answers!
> > I agree that self-concordant functions may no have Lipschitz continuous gradient, and so your example makes sense to me.
> > I think it would be great to add these clarifications and examples into the final version of your paper.
> >
> > I am happy to increase my score.

---

> > > ### Author Response · Authors · 2021-08-24
> > > **Addition of clarifications/examples to revision**
> > >
> > > Thank you for your feedback. We agree and we will make sure to emphasize this in the revision of our paper.

---

### Official Review · Reviewer_62NX · 2021-07-10

**Rating:** 6
**Confidence:** 4

**Summary:**

This work studies simple step size, i.e., 2/(t+2), in FW for generalized self-concordant functions. The main idea is to wait the step size to shrink until f(x_{t+1}) <= f(x_t). By doing so, one can avoid relying on second order information or line search.

**Limitations And Societal Impact:**

This is mainly a theoretical work that helps to understand FW for generalized self-concordant functions.

**Main Review:**

While the main idea is interesting, but it looks to me that waiting for a satisfactory small step size (in Alg.1) is equivalent to a lazy manner for line search. And in fact, while waiting this small step size, one must also check f(x_t) or x_t \in dom f, it is intuitively not convincing that this scheme is more efficient than line search. There are some questions regarding this.

Q1. What are the choices of parameters (e.g., for line search) in different experiments? Is it possible to fine tune them?

Q2. How many iterations are needed to wait for a useful step size? If this number is small, it is not surprising that the proposed algorithm will have a small runtime compared with line-search.

Q3. Suppose that t is large, then it may take a long time to get a satisfactory step size since 2/(t+2) and 2/(t+1+2) may not differ too much. In such a case, it is not clear that why the proposed algorithm is more efficient than line-search.

========
I have read the rebuttal and updated my score accordingly. I do think that more discussions on comparison with line search should be added.


**Time Spent Reviewing:**

4

---

> ### Author Response · Authors · 2021-08-05
> **Response to reviewer 62NX**
>
> We thank the reviewer for the feedback and suggestions, which we believe will help improve the quality of the paper.
> Regarding the specific comments raised by the reviewer:
> 1.	Regarding the comparison of the proposed step size versus line search, we want to highlight an important fact about our work. Note that Algorithm 1 only performs **one gradient evaluation**, **one function evaluation**, **one call to the LP oracle** and **one domain oracle evaluation** to check if $x_{t+1} \notin \mathrm{dom} (f)$, per iteration. This means that Theorem 2.4 effectively bounds the number of zeroth-order, first-order, LP and domain oracle calls needed to achieve a target accuracy $\epsilon$, in terms of $T_{\nu}$ and $\epsilon$; note that $T_{\nu}$ is independent of $\epsilon$. This is an important difference w.r.t. existing bounds, as the existing first-order algorithms for generalized self-concordant functions in the literature that utilize other types of line searches do not bound the number of zeroth-order or the domain oracle calls that are performed in the line search to achieve a target accuracy $\epsilon$. This means that the bounds on these algorithms are typically only informative w.r.t. first-order and LP oracle calls. We will highlight this in a revised version of the paper, providing a table that provides a complete comparison of the oracle complexity of M-FW, versus that of other variants existing in the literature.
> 2.	From the previous comment, one can look at the convergence plots provided in the paper to find the number of iterations in which the M-FW did not “accept” the $2/(t+2)$ step size, as these successive steps would have a constant primal gap value (although it is hard to see with the logarithmic scale in the x-axes). This stands in contrast with the remaining algorithms, which “hide” these extra zeroth-order and domain oracle calls in the convergence plots in terms of iteration count. Regardless of this, in the three experiments that we performed, we did not see the backtracking line search FW variant outperform Algorithm 1 in terms of wall-clock time or iteration count, although we acknowledge that there will of course be instances in which we expect this to happen.
> 3.	Regarding the comment about requiring a high number of iterations until a useful step size is needed, we note that one could search for a suitable step size as follows. Instead of setting $2/(t+1+2)$ when the step size $2/(t+2)$ does not provide progress, we can set $2/(2(t+2))$, thereby halving the step size and performing something akin to binary search for the step size. When taking this into consideration for the convergence proofs, we lose at most a factor of 2 in the convergence bounds shown in the paper, but potentially see a big numerical performance boost in the case where many iterations are needed to provide a useful step size. This numerical enhancement will be added to the paper. Also note that for steps where a step is not accepted neither the point $x_t$, nor the gradient $\nabla f(x_t)$, nor $v_t$ change, so that we do not pay in first-order complexity or LP-complexity when rejecting steps until the first suitable step size is found; we simply query the domain oracle and the zeroth-order oracle.
> 4.	Regarding the parameters used to run the computational experiments, the only parameters that were needed where those for the backtracking line search of Pedregosa et al., and an initial estimate of the smoothness parameter. For the backtracking parameters we simply used the default parameters that were used in the Pedregosa et al. and the Dvurechensky et al. papers. For the initial estimate of the smoothness, we use the suggestion for initialization provided in the Pedregosa et al. paper, which amounts to finding an estimate of the smoothness parameter that holds locally around the optimum. We will specify this in the paper and will also provide an in-depth analysis of the effect of tuning these parameters.

---

### Official Review · Reviewer_JRkk · 2021-07-18

**Rating:** 8
**Confidence:** 5

**Summary:**

Consider the problem of minimizing a generalized self-concordant function on a convex compact set. Notice that a generalized self-concordant function is not smooth (Lipschitz gradient) in general. This paper shows that if we slightly modify the Frank-Wolfe method such that the sequence of function values is non-increasing, then standard analyses of the Frank-Wolfe method based on smoothness and/or strong convexity basically go through. The paper gives a $O(1/t)$ convergence guarantee for the proposed Monotonous Frank-Wolfe (M-FW) method and linear convergence rates for M-FW with backtracking with additional assumptions.

**Limitations And Societal Impact:**

Whereas simplicity in the analysis and algorithm is definitely appreciated, it is natural to ask then, whether there is any tradeoff in the iteration complexity or computational complexity. I suggest the authors make a comparison between their results and results in literature and, if possible, add the discussion to the main text.

**Main Review:**

I like the simple yet very effective idea presented in this paper. Dvurechensky et al., 2020b also study generalized self-concordant minimization by Frank-Wolfe-type methods; the analyses therein treat generalized self-concordant inequalities (Proposition 2.1 in the submission) as analogues of the standard smoothness inequality and require arguably quite complicated calculations; moreover, their analytical step size involves computing the local norm w.r.t. the Hessian of the function, adding computational burden. In comparison, this paper forces the the sequence of function values to be non-increasing. Hence, it suffices to consider the (well-defined) local smoothness and strong convexity parameters on the level set at the initial iterate. The analysis becomes much simpler; the "analytical" step size is simply 2/(t + 2).

The presentation is clear. However, the definition of the Backtrack function in Algorithm 2 is missing in the main text and presented in the appendix, which harms readability.

I have checked the proof in Section 2. I do not check the proofs in the supplementary material.

Other comments:
- As far as I know, the paper "Frank-Wolfe works for non-Lipschitz continuous gradient objectives: Scalable Poisson phase retrieval" by Odor et al. is the first work on Frank-Wolfe minimizing a self-concordant function with a simple step size similar to the one proposed in this paper. Hence, I think the paper should be cited.
- Whereas it is popular to put "all you need" in ML paper titles nowadays, I think such a title is obviously imprecise and mostly misleading. How can one rigorously justify that the step size proposed in this paper is "all you need"?
- Please make $D$ in the Frechet derivatives and $d$ in Remark 2.2 upright. Then, for example, we can easily tell the $D$ in Definition 1.3 is not the diameter of the constraint set.
- I don't get Line 202--205, though the conclusion is obviously correct.
- Line 241: What does "a reasonable assumption" refer to? Perhaps there are missing words.

**Time Spent Reviewing:**

6

---

> ### Author Response · Authors · 2021-08-05
> **Response to reviewer JRkk**
>
> We thank the reviewer for the feedback and suggestions, which we believe will help improve the quality of the paper.
>
> Regarding the specific comments raised by the reviewer:
> 1.	We will try our best to squeeze in the definition of the Backtrack function in the main text, which was left out due to space constraints. We agree with the reviewer, ideally, all the algorithms used in a paper should be included in the main body.
> 2.	We will also cite the paper "Frank-Wolfe works for non-Lipschitz continuous gradient objectives: Scalable Poisson phase retrieval" by Odor et al., as it is indeed an important work that precedes our own. We thank the reviewer for pointing us to the reference.
> 3.	We fully agree that the “X is all you need” format has been (over)-used in the ML community. We were simply trying to convey with a catchy title the fact that the Monotonous Frank-Wolfe algorithm, despite its simplicity and simple proof of convergence, performs admirably in practice, and provides good theoretical guarantees in terms of primal gap and Frank-Wolfe gap.
> 4.	Regarding the use of italicized $D$'s and $d$’s for the derivatives, we agree with the reviewer and will fix these issues in the paper.
> 5.	Regarding the reasoning in Line 202-205: The construction in these lines stems from the fact that the progress bound in Equation 2.4 is only useful in the case where we take a step $x_{t+1} = x_{t} +\gamma_t(v_t – x_t)$, as opposed to the case where $x_{t+1} = x_{t}$. We choose to take a step $x_{t+1} = x_{t} +\gamma_t(v_t – x_t)$ whenever $f(x_{t+1}) \leq f(x_{t})$. The first case (shown in Line 202-205) focuses on a simple condition between the step size $\gamma_t$ and the primal gap $h(x_t)$ which guarantees with Equation 2.4 that $f(x_{t+1}) \leq f(x_{t})$, and ultimately allows us to use Equation 2.4 as a progress bound.
> 6.	Regarding the “reasonable assumption” in Line 241 – we apologize for the imprecision. What we meant to say is that the case in which the optimum is in $\mathrm{Int}(\mathcal{X} \cap \mathrm{dom}(f))$ can happen for example if logarithmic barrier functions are used to encode constraints, and we have that  $\mathrm{dom}(f)$  is a proper subset of $\mathcal{X}$. In this case the optimum is guaranteed to be in $\mathrm{Int}(\mathcal{X} \cap \mathrm{dom}(f))$. We will rephrase this to make it clearer.
> 7.	Regarding the comment about adding an extended computation/iteration complexity comparison (w.r.t. other existing algorithms in the literature): we think this is a great idea that will help highlight the trade-offs between the algorithms. We will incorporate this into our revision of the paper.

---

> > ### Comment · Reviewer_JRkk · 2021-08-22
> > **I keep the rating.**
> >
> > I have read the authors responses. My comments are mainly on the presentation and the authors' responses are satisfactory. I strongly suggest the authors add an iteration/computational complexity comparison with existing results. I think the idea in this paper is already interesting enough to be published, but such a comparison is important to improve the completeness of the paper.
> >
> > @Authors: If possible, could you briefly summarize the comparison with existing results?

---

> > > ### Author Response · Authors · 2021-08-24
> > > **Complexity summary**
> > >
> > >
> > > We happily provide a summary of the complexity comparisons we have already added to the revision of the paper. Note that after our initial submission to NeurIPS, in a recent revision of their work from the 30th of July, the authors of Dvurechensky et al. 2020b added two new second-order algorithms to their paper, namely the MBTFW-GSC and the ASFW-GSC algorithm. We have updated the exposition/comparison, both, in our paper and the summary below to include these two new algorithms.
> > >
> > > Regarding the general complexity guarantees for generalized self-concordant functions over compact convex sets, we note that the number of oracle calls needed to reach an $\epsilon$-accuracy are given by:
> > > 1. FW-GSC [Dvurechensky et al. 2020b, Alg 2]: $\mathcal{O}(1/\epsilon)$ second-order oracle (SOO) calls, $\mathcal{O}(1/\epsilon)$ first-order oracle (FOO) calls and $\mathcal{O}(1/\epsilon)$ linear minimization oracle (LMO) calls.
> > > 2. LBTFW-GSC [Dvurechensky et al. 2020b, Alg 3]: From the theoretical results presented in their work we can derive a complexity of $\mathcal{O}(1/\epsilon)$ FOO calls and $\mathcal{O}(1/\epsilon)$ LMO calls. The authors do not include a complexity bound with respect to zeroth-order oracle (ZOO) calls or domain oracle (DO) calls though, as they focus on iteration complexity. However, one can use Theorem 1 in Appendix C of Pedregosa et al. 2020 to extend the results in their paper, and show a $\mathcal{O}(1/\epsilon)$ ZOO complexity and a $\mathcal{O}(1/\epsilon)$ DO complexity. The constants in both these complexities rely on user-defined initializations.
> > > 3. MBTFW-GSC [Dvurechensky et al. 2020b, Alg 5]: As in the previous algorithm, from the theoretical results in their work we can derive a complexity of $\mathcal{O}(1/\epsilon)$ SOO calls, $\mathcal{O}(1/\epsilon)$ FOO calls, and $\mathcal{O}(1/\epsilon)$ LMO calls. Using Theorem 1 in Appendix C of Pedregosa et al. 2020 one can show again that the algorithm will also need $\mathcal{O}(1/\epsilon)$ ZOO calls and $\mathcal{O}(1/\epsilon)$ DO calls. The constants in both these complexities rely on user-defined initializations as well.
> > > 4. M-FW [This work]: $\mathcal{O}(1/\epsilon)$ FOO calls, ZOO calls, LMO calls and DO calls are needed. These complexities follow inmediately from the iteration-based convergence bounds in the paper, and our algorithm is parameter-free. Moreover, the same complexity bounds hold if we want to arrive at an $\epsilon$-optimal solution in Frank-Wolfe gap (sometimes also called dual gap).
> > >
> > > Regarding the complexity guarantees for generalized self-concordant functions over $(\kappa, q)$-uniformly convex sets, we note that the number of oracle calls that B-FW (which is equivalent to LBTFW-GSC [Dvurechensky et al. 2020b, Alg 3]) needs to reach an $\epsilon$-optimal solution in primal gap is:
> > > 1. If $\mathbf{x}^* \in \mathrm{Int}(\mathcal{X}\cap \mathrm{dom}(f))$ (the optimum is contained in the interior of $\mathcal{X}\cap \mathrm{dom}(f)$) [This work]: $\mathcal{O}(\log 1/\epsilon)$ FOO calls, ZOO calls, LMO calls and DO calls are needed. The constants in the ZOO and DO complexities rely on user-defined initializations.
> > > 2. If $\min_{\mathbf{x}\in \mathcal{X}\cap \mathrm{dom}(f)}\|\| \nabla f(\mathbf{x}) \|\| \geq C > 0$ and $q=2$ [This work]: $\mathcal{O}(\log 1/\epsilon)$ FOO calls, ZOO calls, LMO calls and DO calls are needed. The constants in the ZOO and DO complexities rely on user-defined initializations.
> > > 3. If $\min_{\mathbf{x}\in \mathcal{X}\cap \mathrm{dom}(f)}\| \|\nabla f(\mathbf{x}) \| \|\geq C > 0$ and $q>2$ [This work]: $\mathcal{O}(\epsilon^{-(q-2)/q})$ FOO calls, ZOO calls, LMO calls and DO calls are needed. The constants in the ZOO and DO complexities rely on user-defined initializations.
> > > 4. If there are no straight lines in $\mathcal{X}$ [This work]: $\mathcal{O}(\epsilon^{-(q-1)/q})$ FOO calls, ZOO calls, LMO calls and DO calls are needed. The constants in the ZOO and DO complexities rely on user-defined initialization.
> > >
> > > In the updated version of our paper we have explicitly noted the use of Theorem 1 in Appendix C of Pedregosa et al. 2020 for the DO and ZOO complexity in the four previous cases.
> > >
> > > Regarding the complexity guarantees for generalized self-concordant functions over polytopes we note that the number of oracle calls needed to reach an $\epsilon$-optimal solution in primal gap is:
> > > 1. FW-LLOO [Dvurechensky et al. 2020b, Alg 7]: $\mathcal{O}(\log 1/\epsilon)$ SOO, FOO and LLOO calls. Note that the LMO oracle calls in this algorithm are substituted by **LLOO calls**. The constants in the resulting complexity depend on different geometric parameters (related to the LLOO).
> > > 2. ASFW-GSC [Dvurechensky et al. 2020b, Alg 8]: $\mathcal{O}(\log 1/\epsilon)$ SOO, FOO and LMO calls.
> > > 3. B-AFW [This work]: $\mathcal{O}(\log 1/\epsilon)$ FOO, ZOO, LMO and DO calls. The constants in the ZOO and DO complexities rely on user-defined initialization.
> > >
> > > Note that both the FW-LLOO and ASFW-GSC algorithms are second-order algorithms, whereas the B-AFW is a first-order algorithm.

---

> > > > ### Comment · Reviewer_JRkk · 2021-08-29
> > > > **Keeping my rating**
> > > >
> > > > I was asking for a comparison in terms of the problem parameters. Know results in this direction are all $O(1/\varepsilon)$, unless there are other problem structures to be exploited, so a comparison in terms of the dependence on $\varepsilon$ is not very meaningful... I suggest the authors do a finer comparison.
> > > >
> > > > Anyway, I think the idea in this work is already inspiring enough and hence I keep the rating.

---

### Official Review · Reviewer_pJc9 · 2021-07-20

**Rating:** 8
**Confidence:** 3

**Summary:**

This paper studies the Frank-Wolfe algorithm applied to generalized self-concordant functions. The main result is that the basic Frank-Wolfe method with the standard 2/(t+2) step size and a simple modification to check for feasibility/monotonicty (M-FW) achieves an O(1/t) convergence rate. They show convergence in terms of primal gap, as well as Frank-Wolfe gap, and improved rates under uniform convexity assumptions. Experimental results show that for the class of problems considered, the M-FW method is very competitive, especially in terms of run time, as compared to other FW variants.

**Limitations And Societal Impact:**

Yes/NA

**Main Review:**

Originality:  this work considers a new algorithm/analysis that extends the literature on FW and self-concordant functions. The algorithm proposed is simpler and more elegant than previous algorithms for this problem, and its analysis is clean. The relevant literature is adequately cited and it is clear how this work differs from previous contributions.

Quality:  this is a high quality submission. The results appear to be strong and mathematically correct, to the best of my knowledge. This is a complete and thorough piece of work.

Clarity:  the paper is very well-written and complete.

Significance:  this paper makes a very nice contribution to an important problem. This is the first paper to study the basic Frank-Wolfe method (with a very slight modification) on generalized self-concordant functions. All other papers require stronger assumptions (e.g., second order information) and/or more complicated algorithms. The analysis is clear and the theoretical convergence results are strong. The experimental results also serve to support the theoretical results in a meaningful way. Overall, I think this paper makes a very strong contribution and is perhaps the strongest paper that I've seen on Frank-Wolfe methods specifically for self-concordant functions.

Minor Comments:

1.) The font appears to be non-standard?
2.) Should Theorems 2.9 and 2.10 say "uniformly convex set" instead of "strongly convex set"?

**Time Spent Reviewing:**

2 hours

---

> ### Author Response · Authors · 2021-08-05
> **Response to reviewer pJc9**
>
> We thank the reviewer for the feedback and suggestions, which we believe will help improve the quality of the paper.
>
> Regarding the specific comments raised by the reviewer:
> 1.	Given that this year the NeurIPS guidelines only asked for Type 1, or Embedded TrueType fonts, we decided to use the Charter font in the paper, as we think it provides good readability. We will be happy to switch back to a more standard font if the reviewers believe that it will make the paper easier to read.
> 2.	Regarding Theorem 2.9 and 2.10, the theorems apply to uniformly convex sets. We will fix this in a revised version of the paper and thank the reviewer for spotting this.

---

> > ### Comment · Reviewer_pJc9 · 2021-09-02
> > **response**
> >
> > Thank you for your response. I don't particularly care too strongly about the font, I just wasn't sure if it was allowed as I've never seen it before.

---

### Decision · Program_Chairs · 2021-09-27

**Decision:**

Accept (Poster)

**Comment:**

Overall all reviewers, and also me, liked the paper, which very elegantly simplifies a recent result of interest regarding convergence of Frank-Wolfe for self-concordent functions, which required a more involved appraoch.